## Evidence synthesis

ecology, health and disease and epidemiology

meta-analysis, parasite, prevalence, transmission, urbanization

**Author for correspondence:**
Courtney S. Werner
e-mail: courtney.werner@duke.edu

# Effect of urban habitat use on parasitism in mammals: a meta-analysis

Courtney S. Werner[1] and Charles L. Nunn[1,2]

[1]Department of Evolutionary Anthropology, and [2]Duke Global Health Institute, Duke University, Durham, NC, USA

CSW, 0000-0002-0442-9811; CLN, 0000-0001-9330-2873

Rates of urbanization are increasing globally, with consequences for the dynamics of parasites and their wildlife hosts. A small subset of mammal species have the dietary and behavioural flexibility to survive in urban settings. The changes that characterize urban ecology—including landscape transformation, modified diets and shifts in community composition—can either increase or decrease susceptibility and exposure to parasites. We used a meta-analytic approach to systematically assess differences in endo-parasitism between mammals in urban and non-urban habitats. Parasite prevalence estimates in matched urban and non-urban mammal populations from 33 species were compiled from 46 published studies, and an overall effect of urban habitation on parasitism was derived after controlling for study and parasite genus. Parasite life cycle type and host order were investigated as moderators of the effect sizes. We found that parasites with complex life cycles were less prevalent in urban carnivore and primate populations than in non-urban populations. However, we found no difference in urban and non-urban prevalence for parasites in rodent and marsupial hosts, or differences in prevalence for parasites with simple life cycles in any host taxa. Our findings therefore suggest the disruption of some parasite transmission cycles in the urban ecological community.

## 1. Introduction

Urbanization is transforming patterns of societal growth, resource consumption, energy use and cultural attitudes towards the natural world [1]. More than half of the human population has moved into cities, and the proportion of urban residents is predicted to reach 68% by 2050, with most of the urban growth occurring in lower-income countries [2]. One consequence of urbanization is the rapid expansion of urban areas across the globe, resulting in the transformation of landscapes and the disruption of ecological processes [3]. Over 3% of global land has been modified into an urbanized environment characterized by administrative boundaries, domination by human infrastructure and widespread impervious surface area [4]. Large-scale land-use conversion, in turn, transforms the biogeochemical and climatic processes of an area [5,6] and alters ecological communities [7,8].

Studies that investigate urban ecology find a consistent pattern: urban expansion causes shifts in species composition, which reduce species evenness and often lead to a decrease in biodiversity in urban areas [9,10]. Intense land-use change and urban dangers exclude most native mammals from the urban environment, especially mammals with larger body sizes, slow life histories or specialized diets and activity patterns [11–13]. These 'urban avoiders' are displaced from urbanizing areas by a select few species that establish populations in or around cities [8,10]. Mammal species that survive in urban environments often have omnivorous diets, flexible foraging strategies and increased sociality, though diverse strategies are used to acquire urban space and resources [14]. Some species rely on natural areas surrounding cities but use urban areas for

movement or additional food sources [15]. These 'urban utilizers' include howler monkeys (*Alouatta* spp.) and marmosets (*Callithrix* spp.), known to travel along power lines in order to traverse urban areas, and coyotes (*Canis latrans*) that move through the urban–wildland interface and prey on domestic animals [15–18]. Other species, termed 'urban dwellers', can live entirely within urban boundaries, and include ubiquitous, small-bodied mammals like *Rattus norvegicus* and *Mus musculus* [13,15].

Changes in the ecological structure of urbanized landscapes can have cascading effects on the dynamics of hosts and parasites [19]. Some parasites are expected to disappear from cities along with their native hosts, thus reducing urban parasite biodiversity [20]. However, the urban environment also facilitates unique opportunities for parasite maintenance and transmission, including through interactions between wild mammals, humans and domestic animals [19]. Domestic animals can serve as disease reservoirs, thus increasing disease pressure on wildlife that frequent residential areas [21,22]. In addition, the movement of non-native host species in urban areas may be accompanied by the introduction of novel parasites into urban ecosystems [23].

The health of human populations living in cities is also affected by parasites maintained in urban mammal populations [24,25]. Non-human primates that use urban environments often have contact with humans through food provisioning or conflict [17,26,27]. Due to their close evolutionary relationship to humans, primates are more likely to share parasites with humans [28]. In addition, human populations living in areas of severe urban poverty, where clean water and proper sanitation are inaccessible, are especially vulnerable to zoonotic diseases that further exacerbate the cycle of poverty [29]. Investigating how the urban environment influences host–parasite dynamics is therefore essential for understanding the health of wildlife and humans in cities—including in the context of zoonotic disease transmission—while also informing policy on conservation and human–wildlife conflict management.

Several factors may increase susceptibility or exposure to parasites with different transmission modes in urban relative to non-urban mammal populations. Chemical pollutants and excess artificial light disrupt immune function, rendering urban mammals more vulnerable to infection [30]. Populations of urban mammals may occur at higher densities and forage in areas of clumped food resources, allowing for more frequent interactions with conspecifics that could increase exposure to directly transmitted parasites [9,31–35]. The community composition of the urban environment may exacerbate the transmission of parasites that require intermediate hosts or vectors if a higher relative abundance of urban-dwelling species increases the ratio of competent to non-competent hosts [36]. Finally, contact with humans and human waste may expose urban mammals to zoonotic diseases that are common in humans, particularly in areas without adequate waste disposal [19,29].

Other aspects of the urban environment, however, may reduce parasite risk in urban mammals. Urban wildlife often derive a significant portion of their diet from human foods, are less affected by seasonal food shortages and have better body condition than their counterparts living in non-urban settings [37,38]. Consistent access to higher caloric content may bolster immune function and reduce susceptibility to infection [39]. Similarly, although clumped food resources may facilitate greater contacts for direct parasite transmission

between individuals, the abundance of food may also decrease the need to actively forage, reducing the ranging area and exposure to environmentally transmitted parasites [40]. Finally, many parasites may be absent in the urban environment due to the lack of hosts or proper climatic conditions for their life cycles [20].

A number of studies have investigated the effects of urbanization on parasitism in mammalian wildlife by comparing geographically close populations across anthropogenic disturbance gradients [41–43]. Because these studies occur in vastly different locations and focus on different host and parasite taxa, they often produce different findings. As described above, multiple factors contribute to urban mammal parasite dynamics, yet most pairwise comparisons of urban and non-urban populations lack the statistical power to discern the relative roles of these factors. In addition, the effect of urban habitat on parasitism probably varies across host–parasite systems depending on shifts in the abundance and behaviour of each species in a transmission cycle. A broad, comparative approach enables investigation into the nuanced patterns of interaction across different hosts and parasites in the urban environment.

To more systematically assess the effects of the urban environment on parasitism, we conducted a meta-analysis investigating differences in parasitism between mammals using or dwelling in urban habitats and those living in non-urban habitats. We tested two competing hypotheses concerning the effects of urban habitat use on mammal parasitism. Under the 'urban burden hypothesis', urban habitat use increases parasite prevalence because urban stressors, abundance of competent reservoir species and higher host population density increase susceptibility and exposure to infectious disease. By contrast, the 'urban refuge hypothesis' proposes that urban habitat use decreases mammal parasite prevalence because greater access to resources, less intensive home range use and less suitable habitat for parasites decreases susceptibility and exposure. We predict that parasite prevalence will be higher in urban mammals than in non-urban mammals if the former hypothesis is supported, but the opposite pattern will prevail if the latter hypothesis is supported.

## 2. Methods

### (a) Data collection

We conducted a meta-analysis of studies that reported parasite prevalence data for a mammal population occupying an urban environment and the surrounding non-urban environment. A systematic search of studies published in peer-reviewed journals from 1980 to the present was carried out using the Web of Science 'Topic' function, which scans titles, keywords and abstracts. We initially collected all publications returned after using the search terms 'mammal' AND 'parasit*' AND ('urban* OR 'disturbance' OR 'prevalence'). The term 'disturbance' was included to encompass studies that investigated the effect of human disturbance on parasitism and may have sampled at an urban site. We then collected all additional publications from a second search in which we replaced 'mammal' with 'primat*' OR 'rodent*' OR 'canid*' OR 'felid*' OR 'procyon*' OR 'marsupi*', and 'parasit*' with 'virus*' OR 'bacteria*' OR 'protozoa*' OR 'helminth*'. In addition, we used the primate database within the Global Mammal Parasite Database (GMPD), a source of published parasite data on wild primate populations [44,45], to examine all records for primates in the genera *Papio*, *Cercopithecus*, *Macaca*, *Trachypithecus*, *Semnopithecus*, *Callithrix*, *Sapajus* and

*Alouatta*, which are known to occur in urban areas [46]. We focused on the primate database within the GMPD because these records are most up to date.

The titles and abstracts of returned publications were screened for indications of prevalence estimates in wild mammal populations and narrowed to 433 studies. These studies were scanned further for signs of mammal populations with territories that extend into urban areas, as judged based on terms such as 'city', 'metropolis' and 'high/dense human population', or a map indicating an urban sampling site. We confirmed the original authors' classifications of study sites as urban versus non-urban by using Google Maps to search each sampling location by city name; a site was confirmed as urban if the satellite imagery showed the presence of dominating human infrastructure and artificial substrates. Non-urban areas were characterized by a lower density of humans, an absence of human infrastructure or artificial substrates, and habitat types that included forest reserves, forest fragments and agricultural land.

All prevalence data were extracted from studies that included urban and non-urban mammal populations comprised of the same species and compared within the same country. Publications typically provided prevalence data on multiple parasites, giving multiple effect sizes per study. If a source reported on multiple urban or non-urban populations, the data were averaged by population (weighted by sample size) to calculate a combined prevalence estimate for each habitat type in the study. If a source reported prevalence data along a gradient of urbanization, the individuals sampled in sites with the lowest and highest degrees of urbanization were used to calculate matched prevalence estimates. Zero was included as a reported prevalence if a parasite was searched for but not found in a population. We contacted the corresponding author of a study if the data supporting the study were not entirely reported in the publication but were clearly relevant to our meta-analysis.

A parasite was defined as any micro- or macro-organism that derives benefits from living within a host organism at the expense of the host. All parasites investigated were endoparasites (i.e. living inside the host). The life cycle of each parasite was recorded as complex or simple based on information from the GMPD and the Encyclopaedia of Parasitology [44,45,47]. Parasites with complex life cycles include vector-borne parasites (i.e. transmitted when an infected arthropod bites a competent host) and parasites that require transmission from an intermediate host to a primary host [33]. Parasites with simple life cycles do not require multiple hosts. These include environmentally transmitted parasites that spread through contaminated soil, water or food, and directly transmitted parasites that spread through close contact between conspecifics [33]. The order of each host species was recorded as Carnivora, Primata, Rodentia, Didelphimorphia, Diprotodontia or Permelemorphia. The latter three orders were represented by fewer studies than the others and were combined under the infraclass Marsupialia for analysis.

## (b) Data analysis

An effect size, in the form of a log odds ratio, was obtained for each paired urban and non-urban prevalence comparison. An odds ratio measures the association between an exposure (in this case, to the urban environment) and an outcome (parasite infection). If the log odds ratio is greater than zero, there is a higher prevalence of parasite infection in the exposed, or urban, group and vice versa for a log odds ratio less than zero. If the log odds ratio equals zero, parasite prevalence was the same for both the urban and the non-urban groups. A sampling variance was obtained for each effect size based on the sample size used to estimate prevalence in that study.

A random-effects meta-regression was performed on the effect sizes using the 'rma.mv' function in the 'metafor' package [48] in R.3.6.1. [49], with a significance level of 0.05. Several studies reported prevalence data for more than one parasite in the same population, resulting in multiple effect sizes per study. These data are not independent because individuals that are infected with one parasite may be more susceptible to others; therefore, 'study' was included as a random effect. We also controlled for phylogenetic relationships in our meta-analysis, as traits are often more similar among closely related species [50,51]. The same can be true for effect sizes obtained from different species [52,53]. To account for the phylogeny of host species, a molecular-based phylogeny of mammals [54], along with the 'phytools' and 'geiger' packages [55,56], was used to test the residuals of the random-effects meta-regression model for phylogenetic signal by estimating Pagel's $\lambda$ [57]. The parameter $\lambda$ is estimated using maximum likelihood and ranges from 0 to 1. When $\lambda$ approaches zero, this reduces all internal branches to zero, resulting in a 'star phylogeny' [58], indicating that species are statistically independent of the phylogenetic structure. $\lambda$ retains greater than 90% power for phylogenies with a minimum of 20 species [57]. The detailed phylogenies are not available to assess phylogenetic signal in parasite prevalence; thus, the parasite genus was included as a random effect in the model. Random effects were kept in the model if they had non-zero variance.

After random effects were evaluated, an overall effect of urban habitat on parasite prevalence was obtained. The variables 'host order' and 'parasite life cycle' were then added to the meta-regression model to investigate whether they moderated the overall effect. We also tested for an interaction between host order and parasite life cycle, based on the expectation that variation in urban habitat use may expose host taxa to different types of parasites. In addition, the sampling variances of the effect sizes were investigated as a moderator variable to test for publication bias. If larger sampling variances predict larger effect sizes, this would be consistent with a bias towards publishing smaller studies that report significant differences in parasite prevalence for urban and non-urban habitat types (see [59]). To further investigate publication bias, we applied Egger's test of funnel plot asymmetry to a plot of effect sizes and standard errors [59].

## 3. Results

Our literature search identified 184 effect sizes from 46 published studies in which parasitism was compared between host populations in urban and non-urban settings [41–43,60–102] (electronic supplementary material, S1). In total, parasite prevalence data were available on mammal populations representing 33 species in urban and non-urban habitats of six continents (figure 1a). Seventy-seven unique parasite genera were represented, consisting mostly of helminths (68%), followed by protozoa (23%), bacteria (5%) and viruses (4%). Twelve parasite genera were found in urban wildlife hosts and absent (zero prevalence) from non-urban hosts, while seven genera were found only in non-urban hosts.

Accounting for confounding variables, we found that the urban setting was weakly associated with overall lower parasite prevalence in mammal populations (OR = −0.37, $p = 0.05$; figure 1b). Both 'study' ($\sigma^2 = 0.26$) and 'parasite genus' ($\sigma^2 = 0.52$) were included as random effects. Phylogenetic signal was tested with a phylogeny of 32 out of 33 species; one study reported the prevalence of *Peromyscus leucopus* and *Peromyscus maniculatus* together due to the difficulty of distinguishing between them, so we included *P. leucopus* in the tree for these data [91]. Phylogenetic signal was not evident in residuals from the model (maximum-likelihood estimate of $\lambda = 0$), so no further phylogenetic regression methods were deemed necessary.

Proc. R. Soc. B **287**: 20200397

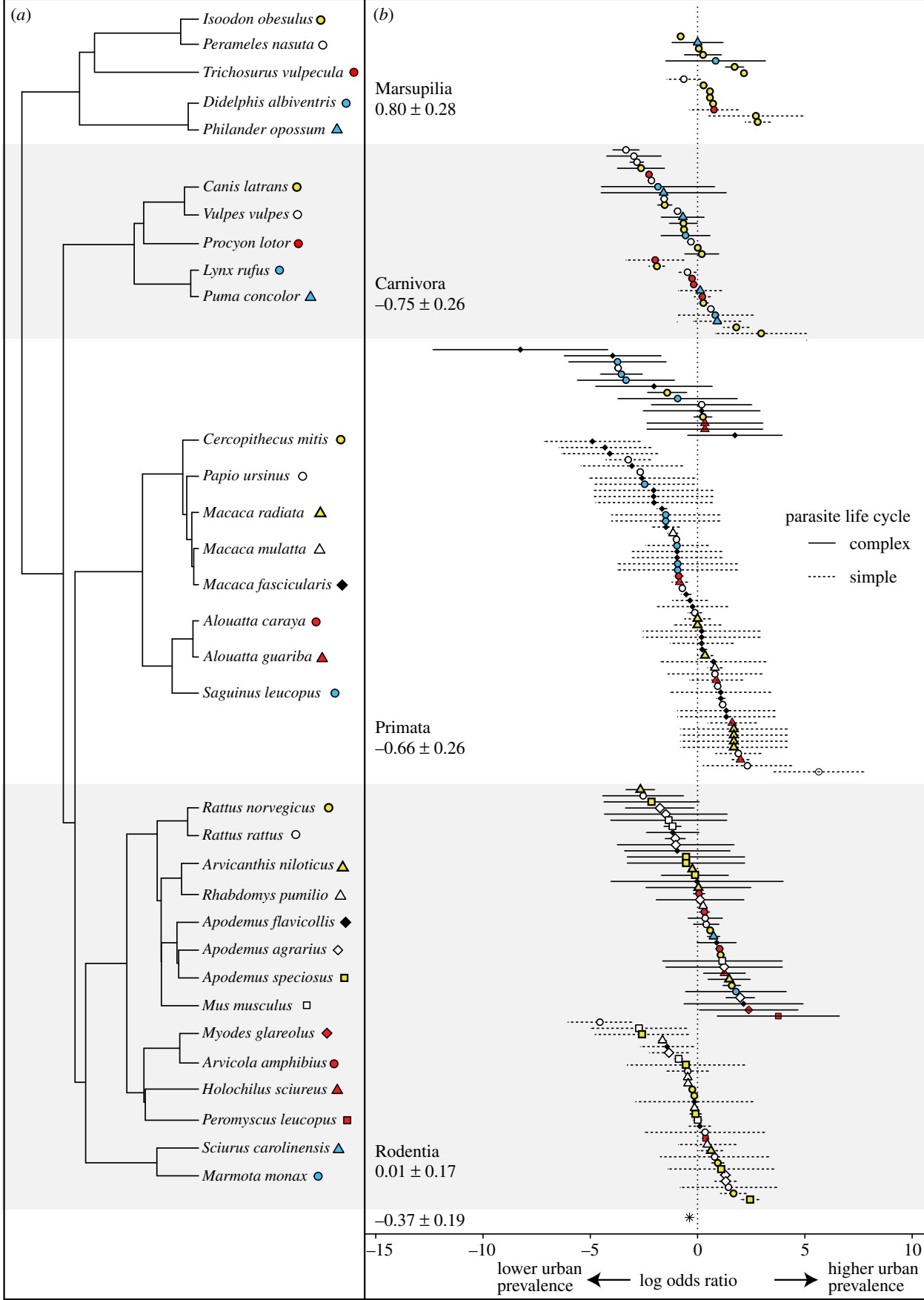

**Figure 1.** Phylogeny and effect sizes. (*a*) Phylogeny of mammal species included in the meta-analysis [53]. Branches are unscaled. The host species within each order are coded by a unique colour/shape combination. (*b*) Forest plot of log odds ratios computed for each host/parasite, arranged by host order and coded by host species according to the phylogeny on the left-hand side. Error bars indicate sampling variance of each effect size, and parasites with complex and simple life cycles are represented by solid and dotted bars, respectively. The mean log odds ratio and standard error are listed for each order. A log odds ratio less than zero indicates lower parasite prevalence in urban habitat relative to non-urban habitat, while a log odds ratio greater than zero indicates a higher prevalence in urban habitat. (Online version in colour.)

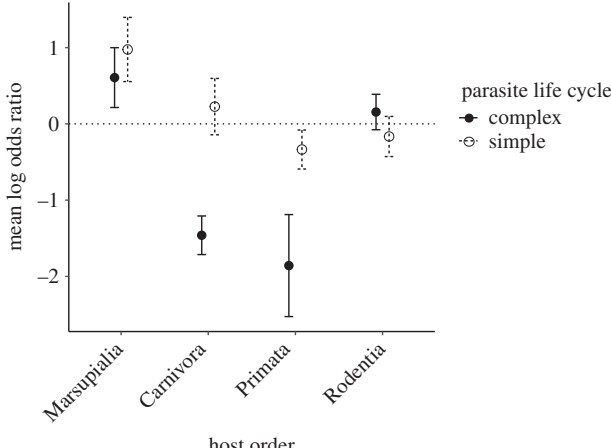

**Figure 2.** Interaction between host order and parasite life cycle moderates parasite prevalence differences. A log odds ratio less than zero indicates a lower prevalence of parasite infection in the urban environment. The urban environment has a more negative effect on the prevalence of complex life cycle parasites, compared to simple life cycle parasites, in Primata and Carnivora, but this trend is not present in Marsupialia or Rodentia. Circles show the mean log odds ratio of each group, while error bars indicate standard errors.

**Table 1.** Coefficients of all variables included in the meta-regression model indicating how the effect of the urban environment on parasite prevalence, calculated as a log odds ratio, is moderated by host order, parasite life cycle and sampling variance of the effect sizes. The effect of 'order' is compared to Marsupialia. An interaction is present between host order and parasite life cycle. A log odds ratio of less than zero indicates a lower prevalence of parasite infection in the urban environment.

| model coefficients | estimate ± s.e. | z-value | p-value |
|---|---|---|---|
| intercept | 0.98 ± 0.50 | 1.98 | 0.05 |
| simple life cycle | −0.01 ± 0.44 | −0.02 | 0.86 |
| Carnivora | −2.48 ± 0.58 | −4.26 | <0.01 |
| Primata | −3.47 ± 0.60 | −5.76 | <0.01 |
| Rodentia | −0.82 ± 0.54 | −1.51 | 0.13 |
| simple × Carnivora | 0.97 ± 0.56 | 1.72 | 0.08 |
| simple × Primata | 2.16 ± 0.53 | 4.04 | <0.01 |
| simple × Rodentia | −0.11 ± 0.50 | −0.21 | 0.83 |
| sampling variance | 0.07 ± 0.10 | 0.65 | 0.51 |

The coefficients from the full meta-regression model revealed variation in effect sizes for the different factors that were included, with some factors (or their interactions) resulting in higher prevalence in urban settings (table 1). Thus, effect sizes were more negative for parasites with complex life cycles in primate hosts ($\mu = -1.86$, $\sigma = 2.59$) and carnivore hosts ($\mu = -1.46$, $\sigma = 1.07$) than for parasites with simple life cycles in primate hosts ($\mu = -0.34$, $\sigma = 1.89$) and carnivore hosts ($\mu = 0.23$, $\sigma = 1.33$), but this interaction between a transmission mode and host taxonomy was not found in marsupials or rodents (figure 2). The significant positive intercept indicates that parasite prevalence was significantly higher in urban settings for Marsupialia. We did not find evidence for publication bias: the sampling variance was not a significant moderator of effect sizes ($\beta = 0.07 \pm 0.10$, $p = 0.51$),

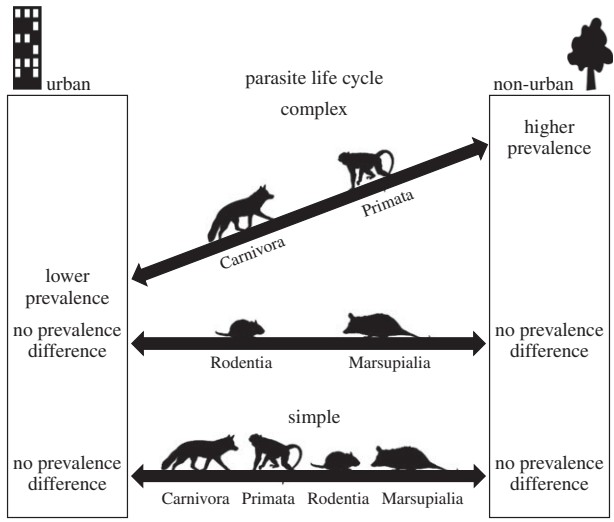

**Figure 3.** Summary of findings. Parasites with complex life cycles were less prevalent in primate and carnivore populations using urban habitats than those in nearby non-urban habitats. No difference in urban and non-urban prevalence was observed for rodent and marsupial hosts, or for parasites with simple life cycles in any of the host taxa. All images are public domain except 'Marsupialia' silhouette, used with permission by Sarah Werning under Creative Commons Attribution 3.0 Unported licence. (https://creativecommons.org/licenses/by/3.0/legalcode).

and the test for funnel plot asymmetry was not significant ($z = -1.30$, $p = 0.19$; electronic supplementary material, S2).

## 4. Discussion

In support of the urban refuge hypothesis, we found that parasites with complex life cycles were less prevalent in primate and carnivore populations using urban habitats than those in surrounding rural or forest habitats. However, no difference in urban and non-urban prevalence was observed for rodent and marsupial hosts, or for parasites with simple life cycles in any of the host taxa (figure 3). The effect of the urban environment on parasite prevalence was variable and highly moderated by the taxonomic order of the host and the life cycle of the parasite, suggesting that unique characteristics of host–parasite systems determine specific responses to highly altered environments. Among papers included in this meta-analysis, observed decreases in urban parasite prevalence were commonly attributed by the authors to a greater availability of human-derived food and elimination of some habitat suitable for parasites or organisms involved in parasite life cycles. On the other hand, observed increases in urban parasite prevalence were attributed by authors to increased host population densities, aggregation of resources and abundance of domestic animals. We consider each of these potential factors in turn.

The consistent availability of food in cities, and resulting shifts in diet, have been well studied in urban wildlife [7]. Altered diets may reduce consumption of intermediate hosts, thereby disrupting the complex life cycles of parasites that infect carnivore and primate definitive hosts. One study included in our analysis found that rural coyotes (*Canis latrans*) consumed more rodents, while urban coyotes foraged on garbage [42]. A similar trend was observed in urban foxes (*Vulpes*

*vulpes*) that were later found to have lower infection rates of *Echinococcus multilocularis*, a helminth that requires rodents as an intermediate host [78,79,81]. Chacma baboons (*Papio ursinus*) are known to forage regularly for human-derived foods in Cape Town, which may reduce consumption of insects [103]. Nematodes of the *Physaloptera* genus depend on beetles or roaches as an intermediate host, and *P. ursinus* closer to Cape Town had a lower prevalence of *Physaloptera* sp. (8%) than a population in a nearby nature reserve (78% [73]).

The climatic and structural changes associated with the urban environment can alter the survival of vectors, intermediate hosts or parasites at various life stages, thereby disrupting the complex life cycles of some parasites. Vector species are small, ectothermic arthropods, and their survival and fitness is sensitive to the higher temperatures of urban air, standing water and physical substrates [104]. Also, the conversion of natural, heterogeneous landscapes to urbanized areas dominated by infrastructure removes favourable breeding habitat for vectors and hosts [105]. For example, *Macracanthorhynchus ingens* primarily infects raccoons (*Procyon lotor*) but uses reptile or amphibian species as paratenic hosts [95]. *Procyon lotor* in rural Canada had a higher rate of *M. ingens* infection than *P. lotor* in urban Ontario, where habitat conditions are less ideal for reptiles and amphibians [95]. Therefore, parasite life cycles dependent on specific food webs are disrupted by not only the overabundance of human food but also the reduced abundance of prey species that are unable to find suitable habitat in the urban environment.

While we found a reduced prevalence of parasites with complex life cycles in carnivore and primate hosts, this pattern was not apparent in rodent and marsupial species. In fact, marsupial hosts seemed to have higher parasite prevalence in the urban environment, although the sample size of studies for this host taxa was smaller than the others. The complex life cycles of certain parasites found in rodents and marsupials may be completed in domestic animals, which are present at high populations in urban areas. For example, domestic cats are definitive hosts of *Toxoplasma gondii* and shed oocysts that infect intermediate hosts [47]. Studies of *T. gondii* in woodchucks (*Marmota monax*) and quenda (*Isoodon obesulus*) suggested that urban prevalence was affected by the increased presence of cats [43,64]. Dogs and cats are also definitive hosts for some cestodes and roundworms identified in urban rodents [79,94].

We found no significant differences in the prevalence of parasites with simple life cycles in urban and non-urban populations for any of the host taxa. Models of host biodiversity loss and parasite species richness have predicted that parasites species with simple life cycles are more robust to decreased host diversity [106]. This anticipated robustness may be reflected in the stable infection rates observed between urban and non-urban mammal populations in our meta-analysis. Individual studies within the meta-analysis did report higher or lower urban prevalence of parasites with simple life cycles. In Australia, for example, *I. obesulus* and *Trichosurus vulpecula* were found to have a higher prevalence of *Giardia* spp. and *Cryptosporidium* spp. [60,64]. Increased urban population densities and aggregation of food and water resources may contribute to an increased likelihood of faecal–oral transmission [64]. On the other hand, *Rhabdomys pumilio* was present at lower population densities in urban environments of the Western Cape Province, South Africa [74]. Reduced nematode prevalence and species richness in

these rodents were attributed to a lack of adequate microhabitat necessary for nematodes to complete the external stages of their life cycles [74]. Variation in effect sizes across studies may also reflect shifting interactions of parasites within hosts in response to urban habitation, especially given the decreased presence of parasites with complex life cycles in some hosts, although further research is needed on this topic [107].

Each of the four mammalian orders included in this meta-analysis are represented by a small subset of species that occupy or use urban environments, indicative of the lack of diversity among urban wildlife. The subfamily Cercopithecinae, for example, was overrepresented among primate species, and Muridae was overrepresented among rodents. Though we did not consider parasite host range as a factor in this study, we predict that the small size of the host community in urban environments may hinder the success of parasites that infect a wide array of hosts in a non-urban setting. Not all urban wildlife was represented; notably, no study on the Virginia opossum (*Didelphis virginiana*) was included, although studies have been done on their diet, home range and body mass in urban areas [108,109]. Overall, this collection of studies represents a reasonably diverse sample of species that use urban environments but excludes the vast majority of fauna native to each region.

In addition, although we sought to include all endoparasite taxa of urban and non-urban mammals, the majority of parasites included were helminths and protozoa. We attribute this to the parasitological methods adopted by most studies, which typically involved visual identification of parasites in the stool or intestinal samples. Additional studies are needed that compare bacterial and viral infections in urban and nonurban wildlife. Furthermore, future research could investigate whether the trends observed in this study apply to multiple measures of parasitism, or whether reduced urban parasite prevalence is associated with increases in other measures, such as individual parasite load.

We did not formally evaluate whether parasites with zoonotic risk to humans are affected differently by the urban environment than non-zoonotic parasites. Many of the studies we included identified parasites to the genus level, making it difficult to determine whether a species was host-specific or to assess its zoonotic status. However, the close proximity of wildlife to areas of dense human activity is likely to increase the risk of zoonotic disease transmission, and risk will be highest where socioeconomic and environmental conditions are appropriate for urban hosts and parasites. For example, inadequate sanitation is associated with an increased presence of rodents in the home, which maintain the cestodes *H. diminuta* and *R. nana* [94,101]. Children of low-income families are at highest risk of infection by these cestodes through accidental ingestion of insects [94].

In conclusion, we found that parasites that require multiple hosts to complete a transmission cycle may decline in urban environments, with some urban wildlife, particularly carnivore and primate species, experiencing a reduction in parasitism. By contrast, parasites that do not rely on multiple species to complete their life cycles survive in the urban environment along with their hosts. The effect of the urban environment on mammal parasitism was variable and highly dependent on the ecological relationships between hosts and parasites. Even in hosts that show declines in parasitism in urban settings, parasites remain present and some pose a risk

to human health in urban settings. As more habitat is transformed into urban landscapes, policy-makers concerned with wildlife and public health in cities should anticipate increased encounters and conflict between humans and urban wildlife and exposure to the parasites they carry. A holistic, sustainable approach to urban planning and policy should account for the interconnectedness of human, animal and environmental health both within and on the expanding edges of cities.

Data accessibility. The dataset and R code supporting this article have been provided as electronic supplementary material.
Competing interests. We declare we have no competing interests.
Funding. We received no funding for this study.

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
