## [Reviewer comments · Proceedings of the Royal Society B: Biological Sciences]

Review History

RSPB-2019-1760.R0 (Original submission)

Review form: Reviewer 1

Recommendation

Accept with minor revision (please list in comments)

Scientific importance: Is the manuscript an original and important contribution to its field?

Good

General interest: Is the paper of sufficient general interest?

Excellent

Quality of the paper: Is the overall quality of the paper suitable?

Good

Is the length of the paper justified?

Yes

Should the paper be seen by a specialist statistical reviewer?

No

Do you have any concerns about statistical analyses in this paper? If so, please specify them explicitly in your report.

No

It is a condition of publication that authors make their supporting data, code and materials available - either as supplementary material or hosted in an external repository. Please rate, if applicable, the supporting data on the following criteria.

Is it accessible?

Yes

Is it clear?

Yes

Is it adequate?

Yes

Do you have any ethical concerns with this paper?

No

Comments to the Author

This was an interesting article with a provoking conclusion, but I have a few comments that I think should be addressed in the intro or discussion.

1. As these evidence synthesis papers are meant to inform policy makers I am concerned that the language is not specific enough given the relatively small number of primate species analyzed in this study. The small sample size is no fault of the authors, they made great effort to include as many as possible, unfortunately, not many met their requirements.

My main concern is that the paper seems to make the conclusion that primates in general have lower prevalence of parasites in urban vs wild habitats when only eight species of primate and two types of parasites (helminths and protozoans) were included in the study. I would argue that these species are another example of urban exploiters and perhaps not representative of the broader population of primates. In the discussion the authors make note of the fact that many primate species cannot utilize urban spaces at all but I think this needs to be expanded on and highlighted sooner, perhaps in the intro, the concern being that the take home message of urban > native habitat could be exploited. To that point, I think it should be made a bit clearer in the introduction and discussion that this study wound up only considering helminths and protozoan parasites. It is unknown what the burden of bacterial and viral parasites is, which I would imagine would be plentiful in an urban landscape (especially human enteric pathogens) and may have changed the results.

2. What is the diversity of parasite in these urban vs wild areas? Is it possible that cityscapes also harbor a reduced diversity of parasites as they do other organisms and that this accounts for some of these results? Do you see the same parasites in the urban species as you do their wild counterparts or do the wild ones not only have more but also different types of parasites?

3. Perhaps address prevalence vs parasite load. It would be interesting to know if the urban primates while harboring fewer types of parasites maintain a higher load of them vs their wild counterparts. Is the parasite able to exploit their host by taking more ground, so to speak? While I know this likely can't be answered with actual parasite load, a statement about how that is something that should be considered would suffice.

4. Lastly, a counter point to the urban refuge argument that clumping of resources = reduced exposure. There is plenty of evidence out there in other species (e.g. prions in cervids) that

clumping of food resources leads to increased rates of transmission. I agree that it could reduce exposure by limiting ranging into new 'contaminated' sites, expenditure of energy, etc. but it could also do just the opposite by congregating susceptible animals in one place to more efficiently spread a parasite.

Just some general curiosity questions that do not need to be addressed, but may be of interest for comparison. I doubt that this can be answered but what is the parasite prevalence in different species of wild populations that exist in similar habitats? As in, do species that are not capable of exploiting urban landscapes have the same prevalence of parasites as species that can live in an urban landscape but do not? Are the urban groups permanent dwellers or do they primarily travel back and forth, and how long have they been in that urban landscape (naïve vs "inoculated" populations)?

Review form: Reviewer 2

Recommendation

Major revision is needed (please make suggestions in comments)

Scientific importance: Is the manuscript an original and important contribution to its field?

Good

General interest: Is the paper of sufficient general interest?

Good

Quality of the paper: Is the overall quality of the paper suitable?

Acceptable

Is the length of the paper justified?

Yes

Should the paper be seen by a specialist statistical reviewer?

Yes

Do you have any concerns about statistical analyses in this paper? If so, please specify them explicitly in your report.

Yes

It is a condition of publication that authors make their supporting data, code and materials available - either as supplementary material or hosted in an external repository. Please rate, if applicable, the supporting data on the following criteria.

Is it accessible?

Yes

Is it clear?

No

Is it adequate?

No

Do you have any ethical concerns with this paper?

No

Comments to the Author

The authors conduct a meta-analysis of the published literature to examine differences in non-human primate parasite prevalence in matched urban and non-urban primate populations. They found 11 studies of 8 species that met their criteria of inclusion. They find that in these 8 species, parasite prevalence was lower in urban primates than in their non-urban counterparts and that this difference was strongest for parasites exhibiting a complex life-cycle; indeed the pattern appears to not be detectable for parasites with a simple life cycle. The authors provide a well-written discussion as to the factors that might be driving these findings.

This represents an important topic and a timely study, but eight species represents a very small number of species for which data are available. One potential solution would be to broaden the scope of their meta-analysis to include mammals more broadly to increase their sample sizes. Given that their predictions are not particularly primate specific, with the exception of the importance of zoonoses, there seems little reason to not include mammals (or vertebrates) more broadly. If the authors would like to control for factors such as phylogeny with this small sample sizes, it would be important for the authors to run a series of simulations to determine what sorts of phylogenetic effects they would in theory be able to detect. If it turns out that this number of species is too small to be able to detect a phylogenetic signal in the data, perhaps this could be left out of the model and this limitation could be discussed in the discussion, though again, increasing the sample size by broadening the taxonomic scope seems like an ideal solution.

The logic driving the author's decision to include 'zoonotic risk to humans (yes or no)' as a predictor variable for differences in parasite prevalence was a bit unclear and could be more clearly explained in the introduction. Further, it appears that there were only 4 parasites/host combinations in the dataset model that are not classified as a zoonosis, likely making it difficult to study the impact of variation in this factor. From the methods, it was unclear whether the specific parasite species detected in these studies were known to be zoonotic to humans, or whether members of the genus are known to be zoonotic? Perhaps the authors could remove this factor from their models and include information on the specific parasite species in their supplementary data table to help authors assess their findings. The introduction could benefit from discussing the fact that many parasites can also move from humans to non-human primates, and perhaps rather than the risk to humans, the emphasis could be on sharing of parasites between humans, domesticated animals, and non-human primates. As the authors mention in the introduction, primate communities in urban landscapes are often species poor, as many species cannot persist in urban landscapes - for those parasites with a broad host range, having fewer other primate species around effectively reduces the population of susceptible hosts. It might be interesting to consider the known host range of these parasites, and if data are available, the primate species richness at these sites.

The manuscript could benefit from further details regarding the data analysis section. For example, the authors state that the appropriate model was chosen by optimizing the AIC; it is unclear how this was done and further, how the significance of specific factors was assessed, the stability of these models, and what model assumptions were tested. Without these details, it is difficult to assess whether this strategy of model selection might in essence be similar to a stepwise approach that runs into a multiple-testing scenario, and it is certainly not possible for someone to replicate their findings given the sparse methods section. The authors could consider sharing their R code with their manuscript (rather than only upon request) and perhaps could consider running only the full model and assessing the significance of factors of interest in this full model as they are aiming to test hypotheses and not prediction. Similarly, while parasite genus is included in the models, there may be too few studies of the same parasite genus (70 effect sizes across, 31 unique genera, across 8 species) to allow an accurate assessment of the impact of this factor. Perhaps it would be beneficial to rather (or additionally) include the broad categories defined above (bacteria, helminth, protozoa, other). On closer examination of the supplementary material, it appears that no parasites in the bacteria or other category were included in the study and perhaps these categories could be removed from the methods.

Minor suggestions:

The introduction jumped between factors that generally may affect animals and their parasites in urban settings and discussions relevant to primates specifically. The text could benefit by restructuring the introduction to provide the theory and more general patterns first, and then narrowing the focus of this manuscript to examples from primates in particular. For example, the paragraph starting on L105 includes many non-primate specific references and could perhaps be merged earlier into the manuscript.

Consider adding details about the number of studies, number of parasites, the taxonomic diversity (helminths and protozoa) and number of primate species to the abstract.

L43: Consider alternative wording for “less developed”.

L88: Missing period at end of sentence.

L100: Consider explaining why these risks are particularly relevant in low and middle-income countries.

L137: This sentence makes it seem like the authors will set out to tackle this problem, as the introduction seems to build to this criticism. Given that this does not end up being the task undertaken, it might be nice to reword to highlight that this is a difficult task that will require much larger sample sizes and resources.

L153: The sudden mention of extreme poverty in the concluding sentence and in the modeling was a bit surprising – consider explaining the logic for why poverty is predicted to influence this relationship more explicitly in the introduction.

Figures: It would be nice to get a sense of the types of parasites that were analyzed in the literature; perhaps Figure 1 could be modified to include information about the parasite species and the primate species as there appears to be a lot of empty space in this figure. What parasites are included in the ‘other’ category? Additionally, a phylogeny of the primate species would be helpful to guide the reader, as would a map of where these studies were conducted (perhaps lines connecting urban and non-urban sites for each species).

Decision letter (RSPB-2019-1760.R0)

27-Aug-2019

Dear Ms Werner:

I am writing to inform you that your manuscript RSPB-2019-1760 entitled "The City as a Refuge from Infectious Disease: A Meta-Analysis of Urban and Non-Urban Primate Parasitism" has, in its current form, been rejected for publication in Proceedings B.

This action has been taken on the advice of referees, who have recommended that substantial revisions are necessary. With this in mind we would be happy to consider a resubmission, provided the comments of the referees are fully addressed. However please note that this is not a provisional acceptance.

The resubmission will be treated as a new manuscript. However, we will approach the same reviewers if they are available and it is deemed appropriate to do so by the Editor. Please note that resubmissions must be submitted within six months of the date of this email. In exceptional

circumstances, extensions may be possible if agreed with the Editorial Office. Manuscripts submitted after this date will be automatically rejected.

Sincerely,
 Evidence Synthesis Editor, Professor Gary Carvalho
 mailto: proceedingsb@royalsociety.org

Comments to Author:

Thank you for submitting your manuscript for consideration as a PRSB, Evidence Synthesis article. It has now been read by 2 experts in the field, and I have also read the manuscript myself. There is a consensus that your manuscript considers a timely and interesting topic, with significant potential. While there are positive comments made, I am unable to take your current manuscript forward. However, I would encourage you to consider a full revision, and a resubmission of your manuscript. As indicated below, I will do my best, to approach the same referees, in the interests of consistency, but cannot of course guarantee their availability. You will see various comments made, and among these, I would like to emphasise the following. First, and I agree with this wholeheartedly, the context and inferences from your manuscript, as currently written, are based on an insufficiently justified sample size. It is necessary, for you to ideally, broaden the taxonomic scope, as referee 2 proposes, or as both referees indicate, a more critical evaluation of the generic nature of your inferences. 2nd, it is really important, especially in our Evidence Synthesis articles, that your manuscript is fully accessible, to a broad readership, including those associated with policy. It is therefore critical, that your data analysis and methodology, is both accessible, and clear. As indicated, currently, your data analysis is not sufficiently clear or detailed, in addition, it is important to provide a clear and accessible exposition of the strategy for adoption of your particular model. I also agree, that your introduction, will require careful restructuring, as suggested. I appreciate that the above represents a major task, that I would encourage you to consider carefully. However, as in all such Resubmissions, I am unable to guarantee eventual publication, but will of course do my best, to expedite the peer review process, aiming for the usual consistency, objectivity, and of course, balance. Finally, please do consider carefully, the extent to which if you proceed with Resubmission, that your manuscript addresses fully, the relevant questions, below, that underpin the structuring of our evidence emphasis articles (Please include, in your uploaded response to referees, a brief response, to each of these questions, some of which, may not be relevant, or fully relevant):

1. Is the key policy-related question(s) articulated clearly?
2. Is there a clear justification in support of policy relevance?
3. Is the likely target audience identified clearly?
4. Does the search for literature utilise a comprehensive range of sources?

5. Does the synthesis article apply clearly documented inclusion criteria to all potentially relevant studies found during the search?
6. Is a clear methodology described to avoid bias?
7. Are study is objectively weighted according to methodological quality of cited literature?
8. Our knowledge gaps and priorities clearly identified?
9. Are outcomes/recommendations tangible in terms of likely impact?
10. Is all necessary supporting information available and accessible

Reviewer(s)' Comments to Author:

Referee: 1

Comments to the Author(s)

This was an interesting article with a provoking conclusion, but I have a few comments that I think should be addressed in the intro or discussion.

1. As these evidence synthesis papers are meant to inform policy makers I am concerned that the language is not specific enough given the relatively small number of primate species analyzed in this study. The small sample size is no fault of the authors, they made great effort to include as many as possible, unfortunately, not many met their requirements.

My main concern is that the paper seems to make the conclusion that primates in general have lower prevalence of parasites in urban vs wild habitats when only eight species of primate and two types of parasites (helminths and protozoans) were included in the study. I would argue that these species are another example of urban exploiters and perhaps not representative of the broader population of primates. In the discussion the authors make note of the fact that many primate species cannot utilize urban spaces at all but I think this needs to be expanded on and highlighted sooner, perhaps in the intro, the concern being that the take home message of urban > native habitat could be exploited. To that point, I think it should be made a bit clearer in the introduction and discussion that this study wound up only considering helminths and protozoan parasites. It is unknown what the burden of bacterial and viral parasites is, which I would imagine would be plentiful in an urban landscape (especially human enteric pathogens) and may have changed the results.

2. What is the diversity of parasite in these urban vs wild areas? Is it possible that cityscapes also harbor a reduced diversity of parasites as they do other organisms and that this accounts for some of these results? Do you see the same parasites in the urban species as you do their wild counterparts or do the wild ones not only have more but also different types of parasites?

3. Perhaps address prevalence vs parasite load. It would be interesting to know if the urban primates while harboring fewer types of parasites maintain a higher load of them vs their wild counterparts. Is the parasite able to exploit their host by taking more ground, so to speak? While I know this likely can't be answered with actual parasite load, a statement about how that is something that should be considered would suffice.

4. Lastly, a counter point to the urban refuge argument that clumping of resources = reduced exposure. There is plenty of evidence out there in other species (e.g. prions in cervids) that clumping of food resources leads to increased rates of transmission. I agree that it could reduce exposure by limiting ranging into new 'contaminated' sites, expenditure of energy, etc. but it could also do just the opposite by congregating susceptible animals in one place to more efficiently spread a parasite.

Just some general curiosity questions that do not need to be addressed, but may be of interest for comparison. I doubt that this can be answered but what is the parasite prevalence in different species of wild populations that exist in similar habitats? As in, do species that are not capable of

exploiting urban landscapes have the same prevalence of parasites as species that can live in an urban landscape but do not? Are the urban groups permanent dwellers or do they primarily travel back and forth, and how long have they been in that urban landscape (naïve vs “inoculated” populations)?

Referee: 2

Comments to the Author(s)

The authors conduct a meta-analysis of the published literature to examine differences in non-human primate parasite prevalence in matched urban and non-urban primate populations. They found 11 studies of 8 species that met their criteria of inclusion. They find that in these 8 species, parasite prevalence was lower in urban primates than in their non-urban counterparts and that this difference was strongest for parasites exhibiting a complex life-cycle; indeed the pattern appears to not be detectable for parasites with a simple life cycle. The authors provide a well-written discussion as to the factors that might be driving these findings.

This represents an important topic and a timely study, but eight species represents a very small number of species for which data are available. One potential solution would be to broaden the scope of their meta-analysis to include mammals more broadly to increase their sample sizes. Given that their predictions are not particularly primate specific, with the exception of the importance of zoonoses, there seems little reason to not include mammals (or vertebrates) more broadly. If the authors would like to control for factors such as phylogeny with this small sample sizes, it would be important for the authors to run a series of simulations to determine what sorts of phylogenetic effects they would in theory be able to detect. If it turns out that this number of species is too small to be able to detect a phylogenetic signal in the data, perhaps this could be left out of the model and this limitation could be discussed in the discussion, though again, increasing the sample size by broadening the taxonomic scope seems like an ideal solution.

The logic driving the author's decision to include 'zoonotic risk to humans (yes or no)' as a predictor variable for differences in parasite prevalence was a bit unclear and could be more clearly explained in the introduction. Further, it appears that there were only 4 parasites/host combinations in the dataset model that are not classified as a zoonosis, likely making it difficult to study the impact of variation in this factor. From the methods, it was unclear whether the specific parasite species detected in these studies were known to be zoonotic to humans, or whether members of the genus are known to be zoonotic? Perhaps the authors could remove this factor from their models and include information on the specific parasite species in their supplementary data table to help authors assess their findings. The introduction could benefit from discussing the fact that many parasites can also move from humans to non-human primates, and perhaps rather than the risk to humans, the emphasis could be on sharing of parasites between humans, domesticated animals, and non-human primates. As the authors mention in the introduction, primate communities in urban landscapes are often species poor, as many species cannot persist in urban landscapes – for those parasites with a broad host range, having fewer other primate species around effectively reduces the population of susceptible hosts. It might be interesting to consider the known host range of these parasites, and if data are available, the primate species richness at these sites.

The manuscript could benefit from further details regarding the data analysis section. For example, the authors state that the appropriate model was chosen by optimizing the AIC; it is unclear how this was done and further, how the significance of specific factors was assessed, the stability of these models, and what model assumptions were tested. Without these details, it is difficult to assess whether this strategy of model selection might in essence be similar to a stepwise approach that runs into a multiple-testing scenario, and it is certainly not possible for someone to replicate their findings given the sparse methods section. The authors could consider sharing their R code with their manuscript (rather than only upon request) and perhaps could consider running only the full model and assessing the significance of factors of interest in this

full model as they are aiming to test hypotheses and not prediction. Similarly, while parasite genus is included in the models, there may be too few studies of the same parasite genus (70 effect sizes across, 31 unique genera, across 8 species) to allow an accurate assessment of the impact of this factor. Perhaps it would be beneficial to rather (or additionally) include the broad categories defined above (bacteria, helminth, protozoa, other). On closer examination of the supplementary material, it appears that no parasites in the bacteria or other category were included in the study and perhaps these categories could be removed from the methods.

Minor suggestions:

The introduction jumped between factors that generally may affect animals and their parasites in urban settings and discussions relevant to primates specifically. The text could benefit by restructuring the introduction to provide the theory and more general patterns first, and then narrowing the focus of this manuscript to examples from primates in particular. For example, the paragraph starting on L105 includes many non-primate specific references and could perhaps be merged earlier into the manuscript.

Consider adding details about the number of studies, number of parasites, the taxonomic diversity (helminths and protozoa) and number of primate species to the abstract.

L43: Consider alternative wording for “less developed”.

L88: Missing period at end of sentence.

L100: Consider explaining why these risks are particularly relevant in low and middle-income countries.

L137: This sentence makes it seem like the authors will set out to tackle this problem, as the introduction seems to build to this criticism. Given that this does not end up being the task undertaken, it might be nice to reword to highlight that this is a difficult task that will require much larger sample sizes and resources.

L153: The sudden mention of extreme poverty in the concluding sentence and in the modeling was a bit surprising – consider explaining the logic for why poverty is predicted to influence this relationship more explicitly in the introduction.

Figures: It would be nice to get a sense of the types of parasites that were analyzed in the literature; perhaps Figure 1 could be modified to include information about the parasite species and the primate species as there appears to be a lot of empty space in this figure. What parasites are included in the ‘other’ category? Additionally, a phylogeny of the primate species would be helpful to guide the reader, as would a map of where these studies were conducted (perhaps lines connecting urban and non-urban sites for each species).

Author's Response to Decision Letter for (RSPB-2019-1760.R0)

See Appendix A.

RSPB-2020-0397.R0

Review form: Reviewer 1

Recommendation

Accept as is

Scientific importance: Is the manuscript an original and important contribution to its field?

Good

General interest: Is the paper of sufficient general interest?

Good

Quality of the paper: Is the overall quality of the paper suitable?

Good

Is the length of the paper justified?

Yes

Should the paper be seen by a specialist statistical reviewer?

No

Do you have any concerns about statistical analyses in this paper? If so, please specify them explicitly in your report.

No

It is a condition of publication that authors make their supporting data, code and materials available - either as supplementary material or hosted in an external repository. Please rate, if applicable, the supporting data on the following criteria.

Is it accessible?

Yes

Is it clear?

Yes

Is it adequate?

Yes

Do you have any ethical concerns with this paper?

No

Comments to the Author

This is a much improved and expanded version of the previous manuscript. The distinction between urban exploiters and urban avoiders puts everything in a better context. There is a lot of synthesizing and categorizing the data in the discussion which was great, but I struggled to visualize it in the results; it might be nice to have a summary figure of the takeaway message. I've attached a very rough sketch (that you can take or leave) of how I saw it in my head. Overall, I think the authors did a nice job of bringing together a lot of studies that will hopefully spur more detailed (e.g. beyond genus ID) work from researchers in the future.

An editorial note: There is a font change from the end of line 224 through the end of 226.

Review form: Reviewer 2 (Jan F. Gogarten)

Recommendation

Accept with minor revision (please list in comments)

Scientific importance: Is the manuscript an original and important contribution to its field?

Excellent

General interest: Is the paper of sufficient general interest?

Excellent

Quality of the paper: Is the overall quality of the paper suitable?

Excellent

Is the length of the paper justified?

Yes

Should the paper be seen by a specialist statistical reviewer?

No

Do you have any concerns about statistical analyses in this paper? If so, please specify them explicitly in your report.

No

It is a condition of publication that authors make their supporting data, code and materials available - either as supplementary material or hosted in an external repository. Please rate, if applicable, the supporting data on the following criteria.

Is it accessible?

Yes

Is it clear?

Yes

Is it adequate?

Yes

Do you have any ethical concerns with this paper?

No

Comments to the Author

I congratulate the authors on a thorough revision and greatly expanding their dataset. Through expand the scope of their meta-analysis their manuscript has greatly improved and their findings are interesting and compelling. I also congratulate the authors on embracing reproducible and transparent science through their sharing of the raw data as well as their R code (for executing their code, the file "output.nex" that contains the host phylogeny could be provided to make life easier for those seeking to reproduce their findings). The authors now present an analysis of 31 host species studied through 42 different studies, which represent 177 effect sizes for parasites in different hosts spanning much of the mammal phylogeny. The larger sample size allows the authors to uncover an interesting and complicated set of interactions; for primates and carnivores, parasites with a complex life cycle are less prevalent in urban areas than in rural/forest areas. In contrast, for parasites with a simple life cycle, there is little/no difference between urban and rural/forest populations in any taxonomic group of hosts. This finding is very much in line with the literature on reductions of biodiversity in urban environments and

seems to represent an important contribution to the field. I would encourage the authors to bring this finding more to the forefront in their abstract, introduction, and discussion.

For example, in the abstract the authors say that parasite prevalence was lower in urban than non-urban mammals, but given the different patterns based on the host order and life cycle of the parasites, such a sweeping statement may not be warranted or helpful in interpreting the data. Indeed, had the literature sampled more marsupial hosts/parasites then this global effect might have disappeared, but this would not at all diminish the important finding about parasites with complex life cycles in primates and carnivores. Rather than putting this statement (and p-value) into the abstract, I would urge the authors to remove the speculation about why these patterns might be occurring (L33-36) and rather use the remainder of the abstract to detail their very interesting findings about life cycles of parasites and specific groups of hosts. In addition, I think it is striking that there is a lot of variation in log odds ratios for these different studies/hosts/parasites and the authors could highlight this finding as well; there may be many interesting factors about these parasite/host combinations that explain this extensive variation. While beyond the scope of this meta-analysis, it suggests further research is needed explore the mechanisms underlying this variation (e.g., how are the interactions between parasites within-hosts impacted by the reduction of parasites with a complex life cycle in urban environments).

One question that arose while reading the text, which I wonder if the authors could elaborate on, is their selection of search criteria. This seems to be a critical part of their methods and in particular, the use of the search term 'disturbance' was a bit puzzling as it wasn't mentioned in the hypotheses and the logic for its inclusion was not immediately clear; in theory one could imagine abstracts comparing viruses of rural and urban rodents that need not mention 'disturbance' or 'parasite' at all. Did all of the search terms on L136 have to be present, or were papers that did not have the word 'disturbance' also included in their literature search? It seems likely the authors explored several search term combinations to maximize the papers found and then settled on the most effective ones; perhaps the authors could simply describe their selection process/logic or explore these search parameters a bit more systematically in the text? For example for viruses, is 'parasite*' always included in the keywords, abstracts, or titles of papers that compare viral populations in mammal populations in rural and urban environments (e.g., while not comparing rural and urban populations explicitly, the paper "Non-random patterns in viral diversity" by Anthony et al., Nature Communications 2015 does not have the word parasite anywhere in the manuscript) and I wonder if the emphasis on parasites may have missed papers focusing on viruses or bacteria. It is certainly possible that efforts presented here have found most of the relevant papers on parasites of mammals in rural and urban settings, but by explaining their selection of search terms and being a bit clearer about the methods of the search, they might be able to convince the readers that this is the case. As a reality check, perhaps the authors could replace parasite with the search terms, "virus*", "bacteria*", "protozoa*", or "helminth*", and remove the word "disturbance" from their search terms to see whether any additional publications are discovered? Perhaps this sort of approach was what the authors were doing with their "genus-specific searches" (L139-141), but what exactly was done was a bit unclear with the limited description provided and it did not seem to be sufficiently explained for an independent reproduction of their efforts. A more thorough and clear description of how papers were collected might be helpful.

Figure 1A: At least in the acrobat version used by the reviewer, the phylogeny seems to have been rendered with some issues – there is a vertical line to the left of the phylogeny that should be removed and the branches do not seem to connect in all cases; there are also some extra lines on some of the branches and there are some pieces of the label names that seem to have been over written that are still showing. Perhaps this was a problem in combining the two figures or is just a problem for the version of acrobat used by the reviewer (in which case please ignore this comment); it might be worth a double check that this figure is reproducing as planned. A scale bar for the host phylogeny might be useful to add as well, to give the reader a sense of the deep evolutionary scales that were explored with this rich dataset.

Figure 1B: It seems like it would be possible to indicate the host species in this figure; for example, the authors could color the host species for each group with a different color in A, and then color the log odds ratios in B accordingly. Alternatively, each host species could get a unique letter (or a unique capital or lowercase letter) or shape (e.g., a silhouette of the host), and this could be placed at the mean log odds ratio. In B, there are also some squares in some of the standard error bars (looks like they are at the mean log odds ratio) – this is also mentioned in the figure legend but I believe is no longer plotted for all parasites. This might be something to double check as it seems they should either be there for all of the log odds ratios, or removed for all. What would also be very helpful is if the authors could show the parasite life cycles in this figure at the same time as the host; it seems to be such a neat aspect of their study. While this potentially represents a lot of information, it could be done simply with the use of dashed or solid lines, or the use of capital and lowercase letters if the strategy mentioned above is used. The other option would be to create two of these forest plots, one for parasites with a simple life cycle, and one with those with a complex life cycle, which might improve interpretability. Given their striking finding regarding the impact of parasite life cycles on these log odds ratios, incorporating this information into this plot seems like it could be very helpful for the reader to understand the dataset. Perhaps in the figure, the authors could indicate what negative and positive values mean (i.e., higher parasite risk in rural areas <- -> higher parasite risk in urban areas).

Minor comments and suggestions:

Keywords: I wonder if epidemiology and zoonoses are appropriate key words given the changes the authors have made. Consider replacing.

L21: Perhaps “potentially impact” rather than “profound consequences”, as that seems to be one the knowledge gap this article is addressing.

L23: Perhaps rather than “use resources”, simply “survive” or “thrive”; there are potentially many factors that limit the ability of a species to thrive in an environment, not simply the use of resources.

L40: Is this rate of expansion unprecedented in human history? On what scale (decades, years, centuries, millennia) and what metric do the authors have in mind (%land use, % of human population living in cities)? Consider rewording (e.g., urban areas are expanding and transforming landscapes) or clarifying what is unprecedented.

L54-56: I appreciate the sentiment, but I would encourage the authors to not put too negative a spin on urbanization, at least in terms of wildlife conservation. Indeed, there are many that argue that urbanization represents one way that humans could reduce their impact on wildlife populations; humans living in rural environments often are not leaving space for wildlife either (or humans only tolerate some kinds of wildlife in this rural areas), and a lot of close contact with wildlife (e.g., hunting) takes place that likely does not take place so much in urban settings. In addition, parasites are only one metric of an individual or an animal populations health or viability and parasites per say do not necessarily indicate a struggling individual or population. Given that only 3% of global land area falls into the category of ‘urban’, I think statements that urbanization is a rapid mechanisms of habitat loss leading to population decline are a bit extreme. I would encourage the authors to take a more nuanced perspective here and throughout the manuscript, on how urbanization might impact animal populations. This need not detract from their findings, which are striking and interesting. If people are leaving rural areas to live in cities, and these rural areas can, at least in part, be returned to wildlife, this could be a very efficient way to protect wildlife and reduce the human impact on wildlife populations. In addition, cities can be much more efficient for many services etc., so their environmental impact (e.g., carbon emissions) per capita may be less than for people living in rural areas. All that to say, the impact of urbanization on wildlife conservation is likely complicated.

L68: Consider removing the word profound and highlighting that this isn’t always the case: “can” might be helpful here.

L91: Consider adding a “can” or “may”

L139-141: As discussed above, it was a bit unclear what was being done with these genus specific searches or how one might do this to replicate the study; please clarify. Should the database now be the GMPD rather than GPPD given the emphasis now on mammals?

L307: Perhaps clarify with another word than taxa (could even list the bigger taxonomic groups meant)?

L338: Did the authors confirm that these mammals are the definitive host for these parasites? It seemed a bit surprising that definitive was not mentioned in the intro, methods or results, but features in the discussion/conclusion; perhaps the authors could consider clarifying in the methods/results/discussion how this concept was brought into the analysis and interpretation or leave out this concept in the discussion and focus rather simply on complex vs. simple life cycles to simplify interpretation.

L338-340: Given that the authors haven't looked at food abundance here and it hard to know how these parasites impact fitness and survival etc., this seems speculative. Consider removing here, and sticking to the really interesting findings uncovered through this meta-analysis.

Decision letter (RSPB-2020-0397.R0)

12-Mar-2020

Dear Ms Werner:

Your manuscript has now been peer reviewed and the reviews have been assessed by an Associate Editor. The reviewers' comments (not including confidential comments to the Editor) and the comments from the Associate Editor are included at the end of this email for your reference. As you will see, the reviewers and the Editors have raised some concerns with your manuscript and we would like to invite you to revise your manuscript to address them.

Research ethics:

Use of animals and field studies:

Please submit a copy of your revised paper within three weeks. If we do not hear from you within this time your manuscript will be rejected. If you are unable to meet this deadline please let us know as soon as possible, as we may be able to grant a short extension.

Best wishes,
Professor Carvalho
Editor, Proceedings B
mailto: proceedingsb@royalsociety.org

Comments to Author:

Thank you very much for the resubmitted version of your manuscript. You will be pleased to see that both referees and I acknowledge the significant improvement in quality and robustness in the coverage of the literature, and inferences contained therein. Your manuscript is now much more in line with the collective criteria underpinning our Evidence Synthesis articles, and we appreciate your constructive responses to suggestions made. As you will see, there remain some issues, that I would ask you to consider. In addition to some general comments on presentation and the quality of figures, I emphasise in particular the following. I agree that the finding that with parasites that have simple life histories, and that show no substantive differences between urban and rural populations in any taxonomic group, is a major finding that is in line with various predictions. Therefore, additional emphasis of this would be helpful. I also agree with the first referee that the generalisation in relation to parasite prevalence and differences between urban and nonurban areas, is oversimplistic, and requires additional critique and balance. Further details are included in the comments below. Finally, and this is especially important in Evidence Synthesis articles, we do require further information and/or clarity, in the selection of your search criteria. As you know, we aim to provide a representative and robust synthesis and critique of the questions being addressed, and it is vitally important, that the reader and any inferences in relation to policy, have appropriate confidence in the patterns emerging. Again, further details are provided below. It is always helpful, as proposed by the 2nd referee, to communicate your key findings in relation to the context of policy, in an easily accessible schematic. The 2nd referee has kindly attached an example of such a summary, and I would encourage you to explore and include something similar.

It is likely that, depending upon availability, I will request a review from one of the original referees, and would ask you to do your best in terms of helping the remaining peer review process, to be as efficient as possible. Therefore, as indicated in the decision letter, please respond to the key issues point by point, with an appropriate justification. We very much look forward to receiving your revised manuscript as soon as you are able.

Reviewer(s)' Comments to Author:

Referee: 2

Comments to the Author(s).

I congratulate the authors on a thorough revision and greatly expanding their dataset. Through expand the scope of their meta-analysis their manuscript has greatly improved and their findings are interesting and compelling. I also congratulate the authors on embracing reproducible and transparent science through their sharing of the raw data as well as their R code (for executing their code, the file "output.nex" that contains the host phylogeny could be provided to make life easier for those seeking to reproduce their findings). The authors now present an analysis of 31 host species studied through 42 different studies, which represent 177 effect sizes for parasites in different hosts spanning much of the mammal phylogeny. The larger sample size allows the authors to uncover an interesting and complicated set of interactions; for primates and carnivores, parasites with a complex life cycle are less prevalent in urban areas than in rural/forest areas. In contrast, for parasites with a simple life cycle, there is little/no difference between urban and rural/forest populations in any taxonomic group of hosts. This finding is very much in line with the literature on reductions of biodiversity in urban environments and seems to represent an important contribution to the field. I would encourage the authors to bring this finding more to the forefront in their abstract, introduction, and discussion.

For example, in the abstract the authors say that parasite prevalence was lower in urban than non-urban mammals, but given the different patterns based on the host order and life cycle of the parasites, such a sweeping statement may not be warranted or helpful in interpreting the data. Indeed, had the literature sampled more marsupial hosts/parasites then this global effect might have disappeared, but this would not at all diminish the important finding about parasites with complex life cycles in primates and carnivores. Rather than putting this statement (and p-value) into the abstract, I would urge the authors to remove the speculation about why these patterns

might be occurring (L33-36) and rather use the remainder of the abstract to detail their very interesting findings about life cycles of parasites and specific groups of hosts. In addition, I think it is striking that there is a lot of variation in log odds ratios for these different studies/hosts/parasites and the authors could highlight this finding as well; there may be many interesting factors about these parasite/host combinations that explain this extensive variation. While beyond the scope of this meta-analysis, it suggests further research is needed explore the mechanisms underlying this variation (e.g., how are the interactions between parasites within-hosts impacted by the reduction of parasites with a complex life cycle in urban environments).

One question that arose while reading the text, which I wonder if the authors could elaborate on, is their selection of search criteria. This seems to be a critical part of their methods and in particular, the use of the search term 'disturbance' was a bit puzzling as it wasn't mentioned in the hypotheses and the logic for its inclusion was not immediately clear; in theory one could imagine abstracts comparing viruses of rural and urban rodents that need not mention 'disturbance' or 'parasite' at all. Did all of the search terms on L136 have to be present, or were papers that did not have the word 'disturbance' also included in their literature search? It seems likely the authors explored several search term combinations to maximize the papers found and then settled on the most effective ones; perhaps the authors could simply describe their selection process/logic or explore these search parameters a bit more systematically in the text? For example for viruses, is 'parasit*' always included in the keywords, abstracts, or titles of papers that compare viral populations in mammal populations in rural and urban environments (e.g., while not comparing rural and urban populations explicitly, the paper "Non-random patterns in viral diversity" by Anthony et al., Nature Communications 2015 does not have the word parasite anywhere in the manuscript) and I wonder if the emphasis on parasites may have missed papers focusing on viruses or bacteria. It is certainly possible that efforts presented here have found most of the relevant papers on parasites of mammals in rural and urban settings, but by explaining their selection of search terms and being a bit clearer about the methods of the search, they might be able to convince the readers that this is the case. As a reality check, perhaps the authors could replace parasite with the search terms, "virus*", "bacteria*", "protozoa*", or "helminth*", and remove the word "disturbance" from their search terms to see whether any additional publications are discovered? Perhaps this sort of approach was what the authors were doing with their "genus-specific searchers" (L139-141), but what exactly was done was a bit unclear with the limited description provided and it did not seem to be sufficiently explained for an independent reproduction of their efforts. A more thorough and clear description of how papers were collected might be helpful.

Figure 1A: At least in the acrobat version used by the reviewer, the phylogeny seems to have been rendered with some issues – there is a vertical line to the left of the phylogeny that should be removed and the branches do not seem to connect in all cases; there are also some extra lines on some of the branches and there are some pieces of the label names that seem to have been over written that are still showing. Perhaps this was a problem in combining the two figures or is just a problem for the version of acrobat used by the reviewer (in which case please ignore this comment); it might be worth a double check that this figure is reproducing as planned. A scale bar for the host phylogeny might be useful to add as well, to give the reader a sense of the deep evolutionary scales that were explored with this rich dataset.

Figure 1B: It seems like it would be possible to indicate the host species in this figure; for example, the authors could color the host species for each group with a different color in A, and then color the log odds ratios in B accordingly. Alternatively, each host species could get a unique letter (or a unique capital or lowercase letter) or shape (e.g., a silhouette of the host), and this could be placed at the mean log odds ratio. In B, there are also some squares in some of the standard error bars (looks like they are at the mean log odds ratio) – this is also mentioned in the figure legend but I believe is no longer plotted for all parasites. This might be something to double check as it seems they should either be there for all of the log odds ratios, or removed for all. What would also be very helpful is if the authors could show the parasite life cycles in this figure at the same time as the host; it seems to be such a neat aspect of their study. While this

potentially represents a lot of information, it could be done simply with the use of dashed or solid lines, or the use of capital and lowercase letters if the strategy mentioned above is used. The other option would be to create two of these forest plots, one for parasites with a simple life cycle, and one with those with a complex life cycle, which might improve interpretability. Given their striking finding regarding the impact of parasite life cycles on these log odds ratios, incorporating this information into this plot seems like it could be very helpful for the reader to understand the dataset. Perhaps in the figure, the authors could indicate what negative and positive values mean (i.e., higher parasite risk in rural areas <- -> higher parasite risk in urban areas).

Minor comments and suggestions:

Keywords: I wonder if epidemiology and zoonoses are appropriate key words given the changes the authors have made. Consider replacing.

L21: Perhaps “potentially impact” rather than “profound consequences”, as that seems to be one the knowledge gap this article is addressing.

L23: Perhaps rather than “use resources”, simply “survive” or “thrive”; there are potentially many factors that limit the ability of a species to thrive in an environment, not simply the use of resources.

L40: Is this rate of expansion unprecedented in human history? On what scale (decades, years, centuries, millennia) and what metric do the authors have in mind (%land use, % of human population living in cities)? Consider rewording (e.g., urban areas are expanding and transforming landscapes) or clarifying what is unprecedented.

L54-56: I appreciate the sentiment, but I would encourage the authors to not put too negative a spin on urbanization, at least in terms of wildlife conservation. Indeed, there are many that argue that urbanization represents one way that humans could reduce their impact on wildlife populations; humans living in rural environments often are not leaving space for wildlife either (or humans only tolerate some kinds of wildlife in this rural areas), and a lot of close contact with wildlife (e.g., hunting) takes place that likely does not take place so much in urban settings. In addition, parasites are only one metric of an individual or an animal populations health or viability and parasites per say do not necessarily indicate a struggling individual or population. Given that only 3% of global land area falls into the category of ‘urban’, I think statements that urbanization is a rapid mechanisms of habitat loss leading to population decline are a bit extreme. I would encourage the authors to take a more nuanced perspective here and throughout the manuscript, on how urbanization might impact animal populations. This need not detract from their findings, which are striking and interesting. If people are leaving rural areas to live in cities, and these rural areas can, at least in part, be returned to wildlife, this could be a very efficient way to protect wildlife and reduce the human impact on wildlife populations. In addition, cities can be much more efficient for many services etc., so their environmental impact (e.g., carbon emissions) per capita may be less than for people living in rural areas. All that to say, the impact of urbanization on wildlife conservation is likely complicated.

L68: Consider removing the word profound and highlighting that this isn’t always the case: “can” might be helpful here.

L91: Consider adding a “can” or “may”

L139-141: As discussed above, it was a bit unclear what was being done with these genus specific searches or how one might do this to replicate the study; please clarify. Should the database now be the GMPD rather than GPPD given the emphasis now on mammals?

L307: Perhaps clarify with another word than taxa (could even list the bigger taxonomic groups meant)?

L338: Did the authors confirm that these mammals are the definitive host for these parasites? It seemed a bit surprising that definitive was not mentioned in the intro, methods or results, but features in the discussion/conclusion; perhaps the authors could consider clarifying in the methods/results/discussion how this concept was brought into the analysis and interpretation or leave out this concept in the discussion and focus rather simply on complex vs. simple life cycles to simplify interpretation.

L338-340: Given that the authors haven’t looked at food abundance here and it hard to know how these parasites impact fitness and survival etc., this seems speculative. Consider removing here, and sticking to the really interesting findings uncovered through this meta-analysis.

Referee: 1

Comments to the Author(s).

This is a much improved and expanded version of the previous manuscript. The distinction between urban exploiters and urban avoiders puts everything in a better context. There is a lot of synthesizing and categorizing the data in the discussion which was great, but I struggled to visualize it in the results; it might be nice to have a summary figure of the takeaway message. I've attached a very rough sketch (that you can take or leave) of how I saw it in my head. Overall, I think the authors did a nice job of bringing together a lot of studies that will hopefully spur more detailed (e.g. beyond genus ID) work from researchers in the future.

An editorial note: There is a font change from the end of line 224 through the end of 226.

Author's Response to Decision Letter for (RSPB-2020-0397.R0)

See Appendix B.

RSPB-2020-0397.R1 (Revision)

Review form: Reviewer 2 (Jan F. Gogarten)

Recommendation

Accept as is

Scientific importance: Is the manuscript an original and important contribution to its field?

Excellent

General interest: Is the paper of sufficient general interest?

Excellent

Quality of the paper: Is the overall quality of the paper suitable?

Excellent

Is the length of the paper justified?

Yes

Should the paper be seen by a specialist statistical reviewer?

No

Do you have any concerns about statistical analyses in this paper? If so, please specify them explicitly in your report.

No

It is a condition of publication that authors make their supporting data, code and materials available - either as supplementary material or hosted in an external repository. Please rate, if applicable, the supporting data on the following criteria.

Is it accessible?

Yes

Is it clear?

Yes

Is it adequate?

Yes

Do you have any ethical concerns with this paper?

No

Comments to the Author

I congratulate the authors for so thoroughly addressing all of the concerns and suggestions that arose through the review process. The modifications have clarified all of my questions and concerns and the findings presented are interesting, compelling, and well presented.

One last minor suggestion; in Figures 1 and 2, the line type (dashed or solid) corresponding to parasites with a simple and complex lifecycle is switched. If it is not a major hassle, it might reduce potential confusion to modify one of the two figures so that the same line type is used for the corresponding life cycle types?

Decision letter (RSPB-2020-0397.R1)

03-Apr-2020

Dear Ms Werner

I am pleased to inform you that your Review manuscript RSPB-2020-0397.R1 entitled "The Effect of Urban Habitat Use on Parasitism in Mammals: A Meta-Analysis" has been accepted for publication in Proceedings B.

The referee(s) do not recommend any further changes. Therefore, please proof-read your manuscript carefully and upload your final files for publication. Because the schedule for publication is very tight, it is a condition of publication that you submit the revised version of your manuscript within 7 days. If you do not think you will be able to meet this date please let me know immediately.

To upload your manuscript, log into <http://mc.manuscriptcentral.com/prsb> and enter your Author Centre, where you will find your manuscript title listed under "Manuscripts with Decisions." Under "Actions," click on "Create a Revision." Your manuscript number has been appended to denote a revision.

You will be unable to make your revisions on the originally submitted version of the manuscript. Instead, upload a new version through your Author Centre.

1) A text file of the manuscript (doc, txt, rtf or tex), including the references, tables (including

captions) and figure captions. Please remove any tracked changes from the text before submission. PDF files are not an accepted format for the "Main Document".

2) A separate electronic file of each figure (tiff, EPS or print-quality PDF preferred). The format should be produced directly from original creation package, or original software format. Please note that PowerPoint files are not accepted.

3) Electronic supplementary material: this should be contained in a separate file from the main text and the file name should contain the author's name and journal name, e.g. `authorname_procb_ESM_figures.pdf`

All supplementary materials accompanying an accepted article will be treated as in their final form. They will be published alongside the paper on the journal website and posted on the online figshare repository. Files on figshare will be made available approximately one week before the accompanying article so that the supplementary material can be attributed a unique DOI. Please see: <https://royalsociety.org/journals/authors/author-guidelines/>

4) Data-Sharing and data citation

It is a condition of publication that data supporting your paper are made available. Data should be made available either in the electronic supplementary material or through an appropriate repository. Details of how to access data should be included in your paper. Please see <https://royalsociety.org/journals/ethics-policies/data-sharing-mining/> for more details.

<http://datadryad.org/submit?journalID=RSPB&manu=RSPB-2020-0397.R1> which will take you to your unique entry in the Dryad repository.

Once again, thank you for submitting your manuscript to Proceedings B and I look forward to receiving your final version. If you have any questions at all, please do not hesitate to get in touch.

Sincerely,

Dr The Proceedings B Team

Associate Editor Board Member: 1

Comments to Author:

Thank you for the thorough revision and for responding so constructively to the PRSB review process throughout. There is one remaining issue that I endorse, to enhance clarity of the Figure. We look forward to seeing this important MS published in the near future.

Reviewer(s)' Comments to Author:

Referee: 2

Comments to the Author(s)

I congratulate the authors for so thoroughly addressing all of the concerns and suggestions that arose through the review process. The modifications have clarified all of my questions and concerns and the findings presented are interesting, compelling, and well presented.

One last minor suggestion; in Figures 1 and 2, the line type (dashed or solid) corresponding to parasites with a simple and complex lifecycle is switched. If it is not a major hassle, it might reduce potential confusion to modify one of the two figures so that the same line type is used for the corresponding life cycle types?

Author's Response to Decision Letter for (RSPB-2020-0397.R1)

See Appendix C.

Decision letter (RSPB-2020-0397.R2)

14-Apr-2020

Dear Ms Werner

I am pleased to inform you that your manuscript entitled "The Effect of Urban Habitat Use on Parasitism in Mammals: A Meta-Analysis" has been accepted for publication in Proceedings B.

Your article has been estimated as being 11 pages long. Our Production Office will be able to confirm the exact length at proof stage.

Open Access

Paper charges

All supplementary materials accompanying an accepted article will be treated as in their final form. They will be published alongside the paper on the journal website and posted on the online

figshare repository. Files on figshare will be made available approximately one week before the accompanying article so that the supplementary material can be attributed a unique DOI.

Sincerely,
Proceedings B
<mailto:proceedingsb@royalsociety.org>

Appendix A

21 February 2020

Dear Professor Carvalho,

Thank you for considering our manuscript, "The City as a Refuge from Infectious Disease: A Meta-Analysis of Urban and Non-Urban Primate Parasitism," for publication in *Proceedings of the Royal Society B*.

We appreciate the feedback from you and the referees. We implemented most of the suggestions, including expanding the analysis to include all mammals. We have accordingly changed the title to, "The Effect of Urban Habitat Use on Parasitism in Mammals: A Meta-Analysis."

Please see below for our responses to the reviewers' suggestions. We have also included a manuscript with tracked changes indicating revisions based on your feedback.

Thank you again for considering our manuscript as an Evidence Synthesis Article in *Proceedings B*. We feel that our manuscript has benefited greatly from the review process, and look forward to hearing from you.

Sincerely,

Courtney S. Werner

Responses to Editor and Reviewers:

Thank you for submitting your manuscript for consideration as a PRSB, Evidence Synthesis article. It has now been read by 2 experts in the field, and I have also read the manuscript myself. There is a consensus that your manuscript considers a timely and interesting topic, with significant potential. While there are positive comments made, I am unable to take your current manuscript forward. However, I would encourage you to consider a full revision, and a resubmission of your manuscript. As indicated below, I will do my best, to approach the same referees, in the interests of consistency, but cannot of course guarantee their availability. You will see various comments made, and among these, I would like to emphasize the following.

First, and I agree with this wholeheartedly, the context and inferences from your manuscript, as currently written, are based on an insufficiently justified sample size. It is necessary, for you to ideally, broaden the taxonomic scope, as referee 2 proposes, or as both referees indicate, a more critical evaluation of the generic nature of your inferences.

As suggested by referee 2, we broadened the taxonomic scope to all mammals, increasing the number of studies from 11 to 42 and the number of effect sizes from 70 to 177. We reframed the meta-analysis as a study of urban-dwelling mammals, and emphasized how this represents a small subset of all mammals.

2nd, it is really important, especially in our Evidence Synthesis articles, that your manuscript is fully accessible, to a broad readership, including those associated with policy. It is therefore critical, that your data analysis and methodology, is both accessible, and clear. As indicated, currently, your data analysis is not sufficiently clear or detailed, in addition, it is important to provide a clear and accessible exposition of the strategy for adoption of your particular model.

We elaborated on the data analysis methods and added a table summarizing our full model.

I also agree, that your introduction, will require careful restructuring, as suggested.

I appreciate that the above represents a major task, that I would encourage you to consider carefully. However, as in all such Resubmissions, I am unable to guarantee eventual publication, but will of course do my best, to expedite the peer review process, aiming for the usual consistency, objectivity, and of course, balance. Finally, please do consider carefully, the extent to which if you proceed with Resubmission, that your manuscript addresses fully, the relevant questions, below, that underpin the structuring of our evidence emphasis articles (Please include, in your uploaded response to referees, a brief response, to each of these questions, some of which, may not be relevant, or fully relevant):

1. Is the key policy-related question(s) articulated clearly?

The key policy related question: How does the urban environment effect parasite load of mammals that utilize or dwell in urban environments? This question is articulated in the final paragraph of the introduction, L124-136

2. Is there a clear justification in support of policy relevance?

See L78-L195. "Investigating how the urban environment influences host-parasite dynamics is therefore essential for understanding the health of wildlife and humans in cities—including in the context of zoonotic disease transmission—while also informing policy on conservation and human-wildlife conflict management"

3. Is the likely target audience identified clearly?

The target audience, other than researchers who will fill in knowledge gaps on this topic, is "policy-makers concerned with wildlife and public health in cities" (L361).

4. Does the search for literature utilise a comprehensive range of sources?

We used Web of Science and the Global Primate Parasite Database to search published literature from 1980 to the present. See L142-L149.

5. Does the synthesis article apply clearly documented inclusion criteria to all potentially relevant studies found during the search?

The inclusion criteria for relevant studies are outlined L154-L172.

6. Is a clear methodology described to avoid bias?

The methods section clearly describes the literature search, inclusion criteria, variable definitions, effect size calculations, and meta-regression model adoption process. We expect that readers could replicate our methods.

7. Are studies objectively weighted according to methodological quality of cited literature?

Studies are weighted based on the sampling variance, which is calculated from the sample size used to calculate prevalence. The variable “study” is a random effect in the model to account for differences in number of effect sizes reported for each study. A sampling variance of each effect size is included in the forest plot (Figure 1).

8. Are knowledge gaps and priorities clearly identified?

The following knowledge gaps and priorities are stated in the paper:

- **The study is interested in urban mammals, which represents a very small subset of all mammals. In addition, this study includes a sample, not a comprehensive list, of all urban mammals. (L323-334)**
- **This study considers mostly helminth and protozoa parasites. Very few viral and bacterial parasites are represented. (L335-339)**
- **The number of studies found for marsupial hosts is smaller than for other taxa. (L304-306)**
- **Determining whether parasites are host-specific or pose zoonotic risk for humans was difficult given the identification methods and novelty of parasites in many of the studies. (L344-L346)**

9. Are outcomes/recommendations tangible in terms of likely impact?

- **See concluding paragraph, L721. The effect of the urban environment on parasitism is specific to certain hosts and parasites. Reduction in urban parasite burden may lead to more urban habitation, more conflict with humans, and more zoonotic risk. Some parasites exist at higher prevalence in urban mammals than others, and some human populations are at greater risk for exposure to parasites from urban wildlife.**

10. Is all necessary supporting information available and accessible

Supporting data and code are both available in supplementary information.

Reviewer(s)' Comments to Author:

Referee: 1

Comments to the Author(s)

This was an interesting article with a provoking conclusion, but I have a few comments that I think should be addressed in the intro or discussion.

1. As these evidence synthesis papers are meant to inform policy makers I am concerned that the language is not specific enough given the relatively small number of primate species analyzed in this study. The small sample size is no fault of the authors, they made great effort to include as many as possible, unfortunately, not many met their requirements.

My main concern is that the paper seems to make the conclusion that primates in general have lower prevalence of parasites in urban vs wild habitats when only eight species of primate and two types of parasites (helminths and protozoans) were included in the study. I would argue that these species are another example of urban exploiters and perhaps not representative of the broader population of primates.

We agree that the scope of the study is limited, and have expanded it to include all mammals (see below). In terms of terminology, we reframed the paper as a meta-analysis of “urban exploiters”, although we altered the language to include the terminology used by Fischer et al. 2015 (“urban utilizers” and “urban dwellers”).

In the discussion the authors make note of the fact that many primate species cannot utilize urban spaces at all but I think this needs to be expanded on and highlighted sooner, perhaps in the intro, the concern being that the take home message of urban > native habitat could be exploited.

L62: We now acknowledge that the majority of mammals fall into the “urban avoider” category, we highlight certain traits that leave these mammals vulnerable to urbanization, and we state conservation consequences. We added a paragraph (L323+) that presents and critically evaluates the scope of species included in this study, and restated how most mammal species are excluded. Finally, we removed the term “refuge” from the title.

To that point, I think it should be made a bit clearer in the introduction and discussion that this study wound up only considering helminths and protozoan parasites. It is unknown what the burden of bacterial and viral parasites is, which I would imagine would be plentiful in an urban landscape (especially human enteric pathogens) and may have changed the results.

In the paragraph beginning L335, we now critically evaluate the limited number of parasite taxa included in this study, and contextualize the results based on these limitations. We also state that further studies are needed to assess patterns relevant to bacterial and viral parasites.

2. What is the diversity of parasite in these urban vs wild areas? Is it possible that cityscapes also harbor a reduced diversity of parasites as they do other organisms and that this accounts for some of these results? Do you see the same parasites in the urban species as you do their wild counterparts or do the wild ones not only have more but also different types of parasites?

A sentence beginning L233 addresses the number of parasite genera found in urban habitat and absent from non-urban habitat, and vice versa. In the discussion, we note that parasites may be absent from urban environments due to lack of host organisms (L289+) or due to microhabitat conditions that are inappropriate for their life cycles (L320+).

3. Perhaps address prevalence vs parasite load. It would be interesting to know if the urban primates while harboring fewer types of parasites maintain a higher load of them vs their wild counterparts. Is the parasite able to exploit their host by taking more ground, so to speak? While I know this likely can't be answered with actual parasite load, a statement about how that is something that should be considered would suffice.

Thank you for this suggestion. Given concerns about the implications and interpretations of variation in parasite load – coupled with the now much expanded sample size and analyses – we opted to retain our focus on parasite prevalence. This also enables us to maximize sample sizes, as fewer studies have compared parasite load in urban and non-urban settings (and also as noted by the reviewer, parasite load itself is rarely measured, i.e. it is proxied with egg counts). However, to address this concern, we now note in the Discussion that future research may wish to also investigate variation in parasite load (L339-342).

4. Lastly, a counter point to the urban refuge argument that clumping of resources = reduced exposure. There is plenty of evidence out there in other species (e.g. prions in cervids) that clumping of food resources leads to increased rates of transmission. I agree that it could reduce exposure by limiting ranging into new 'contaminated' sites, expenditure of energy, etc. but it could also do just the opposite by congregating susceptible animals in one place to more efficiently spread a parasite.

Thank you for making this point. In the Introduction, we now acknowledge the risk of clumped food resources on transmission (99). We repeated this in L112 before introducing the possibility of clumped resources reducing exposure,

thereby presenting both opposing outcomes.

Just some general curiosity questions that do not need to be addressed, but may be of interest for comparison. I doubt that this can be answered but what is the parasite prevalence in different species of wild populations that exist in similar habitats? As in, do species that are not capable of exploiting urban landscapes have the same prevalence of parasites as species that can live in an urban landscape but do not?

Are the urban groups permanent dwellers or do they primarily travel back and forth, and how long have they been in that urban landscape (naïve vs “inoculated” populations)?

Thank you for these thoughtful questions. In relation to the last one, we emphasized that urban wildlife have a variety of strategies for using urban space. In the Introduction, we also clarified the differences between “urban utilizers” and “urban dwellers”(L65+), both of which are included in our analysis.

Referee: 2

Comments to the Author(s)

The authors conduct a meta-analysis of the published literature to examine differences in non-human primate parasite prevalence in matched urban and non-urban primate populations. They found 11 studies of 8 species that met their criteria of inclusion. They find that in these 8 species, parasite prevalence was lower in urban primates than in their non-urban counterparts and that this difference was strongest for parasites exhibiting a complex life-cycle; indeed the pattern appears to not be detectable for parasites with a simple life cycle. The authors provide a well-written discussion as to the factors that might be driving these findings.

This represents an important topic and a timely study, but eight species represents a very small number of species for which data are available. One potential solution would be to broaden the scope of their meta-analysis to include mammals more broadly to increase their sample sizes. Given that their predictions are not particularly primate specific, with the exception of the importance of zoonoses, there seems little reason to not include mammals (or vertebrates) more broadly.

Thank you for this suggestion, which we followed. After broadening the taxonomic scope to all mammals, our sample sizes increased substantially: the number of studies from 11 to 42, the number of effect sizes from 70 to 177, and the number of host species from 8 to 31.

If the authors would like to control for factors such as phylogeny with this small sample sizes, it would be important for the authors to run a series of simulations to determine what sorts of phylogenetic effects they would in theory be able to detect . If it turns out that this number of species is too small to be able to detect a phylogenetic signal in the data, perhaps this could be left out of the model and this limitation could be discussed in the discussion, though again, increasing the sample size by broadening the taxonomic scope seems like an ideal solution.

We referred to the literature for other research that has already investigated statistical power to detect phylogenetic signal based on sample size. Freckleton et al. (2002) addressed this issue in a simulation study. They found that 20 is the minimum number of species needed for 90% power, whereas we included 30 species in our phylogenetic analysis. We now note this in our manuscript (L211).

The logic driving the author's decision to include 'zoonotic risk to humans (yes or no)' as a predictor variable for differences in parasite prevalence was a bit unclear and could be more clearly explained in the introduction. Further, it appears that there were only 4 parasites/host combinations in the dataset model that are not classified as a zoonosis, likely making it difficult to study the impact of variation in this factor. From the methods, it was unclear whether the specific parasite species detected in these studies were known to be zoonotic to humans, or whether members of the genus are known to be zoonotic? Perhaps the authors could remove this factor from their models and include information on the specific parasite species in their supplementary data table to help authors assess their findings.

We agree with your concerns. Identifying which parasites are actually zoonotic is difficult due to the lack of specificity with which most papers reported parasites. Thus, we decided to remove zoonotic risk as a variable. We addressed this limitation in a paragraph on zoonotic risk to humans in the Discussion (L343+).

The introduction could benefit from discussing the fact that many parasites can also move from humans to non-human primates, and perhaps rather than the risk to humans, the emphasis could be on sharing of parasites between humans, domesticated animals, and non-human primates.

We addressed the multidirectional transmission risk in L78. For the paragraph beginning in L85, we maintained the emphasis on risk to humans, as we feel that policy-makers interested in urban public health will be most interested in the implications for human populations.

As the authors mention in the introduction, primate communities in urban landscapes are often species poor, as many species cannot persist in urban landscapes – for those parasites with a broad host range, having fewer other primate species around effectively reduces the population of susceptible hosts. It might be interesting to consider the known host range of these parasites, and if data are available, the primate species richness at these sites.

Thank you for this interesting idea. Although sufficient data are not available on host range for the parasites in our study, we added a point to the Discussion (L327) that the patterns might interact with host range, such that parasites with a broad host range might be expected to drop out of depauperate host communities in urban settings.

The manuscript could benefit from further details regarding the data analysis section. For example, the authors state that the appropriate model was chosen by optimizing the AIC; it is unclear how this was done and further, how the significance of specific factors was assessed, the stability of these models, and what model assumptions were tested. Without these details, it is difficult to assess whether this strategy of model selection might in essence be similar to a stepwise approach that runs into a multiple-testing scenario, and it is certainly not possible for someone to replicate their findings given the sparse methods section. The authors could consider sharing their R code with their manuscript (rather than only upon request) and perhaps could consider running only the full model and assessing the significance of factors of interest in this full model as they are aiming to test hypotheses and not prediction.

Thank you for these suggestions. We ran the full model, reported σ^2 values for random effects, and presented a table of all coefficients (Table 1). We also shared our R code with the manuscript. We expect that readers may now interpret our model in more detail and replicate our findings.

Similarly, while parasite genus is included in the models, there may be too few studies of the same parasite genus (70 effect sizes across, 31 unique genera, across 8 species) to allow an accurate assessment of the impact of this factor. Perhaps it would be beneficial to rather (or additionally) include the broad categories defined above (bacteria, helminth, protozoa, other). On closer examination of the supplementary material, it appears that no parasites in the bacteria or other category were included in the study and perhaps these categories could be removed from the methods.

We believe expanding the scope of the analysis solved this issue. We now have 73 unique genera and 177 effect sizes. In addition, we removed “parasite type” as a factor from the model due to the low numbers of genera in the virus and bacteria category.

Minor suggestions:

The introduction jumped between factors that generally may affect animals and their parasites in urban settings and discussions relevant to primates specifically. The text could benefit by restructuring the introduction to provide the theory and more general patterns first, and then narrowing the focus of this manuscript to examples from primates in particular. For example, the paragraph starting on L105 includes many non-primate specific references and could perhaps be merged earlier into the manuscript.

We are no longer narrowing the scope to only primates; thus, these concerns should be less relevant in the revised manuscript.

Consider adding details about the number of studies, number of parasites, the taxonomic diversity (helminths and protozoa) and number of primate species to the abstract.

We added the number of host species and the number of published studies to the abstract.

L43: Consider alternative wording for “less developed”.

Thank you, we changed this to “lower income.”

L88: Missing period at end of sentence.

Fixed.

L100: Consider explaining why these risks are particularly relevant in low and middle-income countries.

On L89, we elaborated on this.

L137: This sentence makes it seem like the authors will set out to tackle this problem, as the introduction seems to build to this criticism. Given that this does not end up being the task undertaken, it might be nice to reword to highlight that this is a difficult task that will require much larger sample sizes and resources.

The current manuscript makes it clearer that a meta-analytic approach allows us to investigate broader patterns of parasitism in the urban environment, especially with a larger sample size, but that there are still limitations to this approach.

L153: The sudden mention of extreme poverty in the concluding sentence and in the modeling was a bit surprising – consider explaining the logic for why poverty is predicted to influence this relationship more explicitly in the introduction.

We removed “poverty” as a variable from the meta-analysis because the “urban poverty headcount” indicator, reported by the World Bank, is not available for European and North American countries added to this meta-analysis. Suitable

alternative statistics for urban poverty were not found. We still addressed the influence of urban poverty increasing risk of exposure to zoonotic disease (L89, L350).

Figures: It would be nice to get a sense of the types of parasites that were analyzed in the literature; perhaps Figure 1 could be modified to include information about the parasite species and the primate species as there appears to be a lot of empty space in this figure. What parasites are included in the 'other' category? Additionally, a phylogeny of the primate species would be helpful to guide the reader, as would a map of where these studies were conducted (perhaps lines connecting urban and non-urban sites for each species).

Thank you, we decided to combine two of your suggestions and create a figure with a forest plot and phylogeny of hosts side-by-side.

Reference:

Freckleton RP, Harvey PH, Pagel M. 2002 Phylogenetic analysis and comparative data: a test and review of evidence. *The American Naturalist* **160**, 712-726. (doi: [10.1086/343873](https://doi.org/10.1086/343873))

Appendix B

02 April 2020

Dear Editor,

Thank you for considering our manuscript, "The Effect of Urban Habitat Use on Parasitism in Mammals: A Meta-Analysis," for publication in *Proceedings of the Royal Society B*.

We appreciate the feedback from you and the referees, and we have implemented most of the suggestions.

Please see below for our responses to the reviewers' comments. We have also included a manuscript with tracked changes indicating revisions based on your feedback.

Thank you again for considering our manuscript as an Evidence Synthesis Article in *Proceedings B*. We feel that our manuscript has benefited greatly from the review process, and look forward to hearing from you.

Sincerely,

Courtney S. Werner

Comments to Author:

Thank you very much for the resubmitted version of your manuscript. You will be pleased to see that both referees and I acknowledge the significant improvement in quality and robustness in the coverage of the literature, and inferences contained therein. Your manuscript is now much more in line with the collective criteria underpinning our Evidence Synthesis articles, and we appreciate your constructive responses to suggestions made. As you will see, there remain some issues, that I would ask you to consider. In addition to some general comments on presentation and the quality of figures, I emphasise in particular the following. I agree that the finding that with parasites that have simple life histories, and that show no substantive differences between urban and rural populations in any taxonomic group, is a major finding that is in line with various predictions. Therefore, additional emphasis of this would be helpful. I also agree with the first referee that the generalisation in relation to parasite prevalence and differences between urban and nonurban areas, is oversimplistic, and requires additional critique and balance. Further details are included in the comments below.

Thank you very much for the positive feedback. We have altered the Abstract, Introduction, and Discussion to emphasize the nuanced patterns of our findings rather than a broad, oversimplified effect. Please see below for more details.

Finally, and this is especially important in Evidence Synthesis articles, we do require further information and/or clarity, in the selection of your search criteria. As you know, we aim to provide a representative and robust synthesis and critique of the questions being addressed, and it is vitally important, that the reader and any inferences in relation to policy, have appropriate confidence in the patterns emerging. Again, further details are provided below.

We clarified our process for selecting search criteria and specified Boolean Operators. We expect readers may now replicate our search without confusion. Also, per the suggestion of Referee 1, we included new search terms and added four additional publications, which improved the robustness of our study and did not alter our conclusions in any significant way.

It is always helpful, as proposed by the 2nd referee, to communicate your key findings in relation to the context of policy, in an easily accessible schematic. The 2nd referee has kindly attached an example of such a summary, and I would encourage you to explore and include something similar.

We appreciate Referee 2 taking the time to sketch out a schematic, and we used this design as a starting point for a simple figure communicating the key points of our findings.

It is likely that, depending upon availability, I will request a review from one of the original referees, and would ask you to do your best in terms of helping the remaining peer review process, to be as efficient as possible. Therefore, as indicated in the decision letter, please respond to the key issues point by point, with an appropriate justification. We very much look forward to receiving your revised manuscript as soon as you are able.

Reviewer(s)' Comments to Author:

Referee: 2

Comments to the Author(s).

I congratulate the authors on a thorough revision and greatly expanding their dataset. Through expand the scope of their meta-analysis their manuscript has greatly improved and their findings are interesting and compelling. I also congratulate the authors on embracing reproducible and transparent science through their sharing of the raw data as well as their R code (for executing their code, the file "output.nex" that contains the host phylogeny could be provided to make life easier for those seeking to reproduce their findings).

Thank you. When we tried to upload the “output.nex” file to the submission portal as Electronic Supplementary Material, we received an error message that it is not an accepted file type. However, we are happy to provide this file through other means.

The authors now present an analysis of 31 host species studied through 42 different studies, which represent 177 effect sizes for parasites in different hosts spanning much of the mammal phylogeny. The larger sample size allows the authors to uncover an interesting and complicated set of interactions; for primates and carnivores, parasites with a complex life cycle are less prevalent in urban areas than in rural/forest areas. In contrast, for parasites with a simple life cycle, there is little/no difference between urban and rural/forest populations in any taxonomic group of hosts. This finding is very much in line with the literature on reductions of biodiversity in urban environments and seems to represent an important contribution to the field. I would encourage the authors to bring this finding more to the forefront in their abstract, introduction, and discussion.

For example, in the abstract the authors say that parasite prevalence was lower in urban than non-urban mammals, but given the different patterns based on the host order and life cycle of the parasites, such a sweeping statement may not be warranted or helpful in interpreting the data. Indeed, had the literature sampled more marsupial hosts/parasites then this global effect might have disappeared, but this would not at all diminish the important finding about parasites with complex life cycles in primates and carnivores. Rather than putting this statement (and p-value) into the abstract, I would urge the authors to remove the speculation about why these patterns might be occurring (L33-36) and rather use the remainder of the abstract to detail their very interesting findings about life cycles of parasites and specific groups of hosts.

We have made the following changes to emphasize the details of the patterns uncovered by our study, rather than a broad overall effect:

Abstract: We removed the sentence reporting the overall effect size and p-value, and replaced it with details on the effect of urban habitat for each taxonomic group and parasite type, including our finding that prevalence of parasites with simple life cycles do not differ between urban and non-urban habitat.

Introduction: We added two sentences (L114-118) highlighting the importance of a meta-analytic approach for uncovering nuanced patterns in parasitism across host and parasite taxa. In the two paragraphs beginning L86 and L98, we discuss factors that may affect susceptibility and exposure to parasites with different transmission modes—we added a phrase in the topic sentence of L85 to emphasize the distinction between parasite types.

Discussion: The first sentences have been altered to present the main findings, including that lack of prevalence difference in parasites with simple life cycles. Our new figure also highlights this finding. The paragraph in which we discuss parasites with simple life cycles now includes a reference to models from the

literature predicting that simple parasites are more robust than parasites with complex life cycles to reduction in host biodiversity. Our concluding paragraph reiterates the main points of our findings.

In addition, I think it is striking that there is a lot of variation in log odds ratios for these different studies/hosts/parasites and the authors could highlight this finding as well; there may be many interesting factors about these parasite/host combinations that explain this extensive variation. While beyond the scope of this meta-analysis, it suggests further research is needed explore the mechanisms underlying this variation (e.g., how are the interactions between parasites within-hosts impacted by the reduction of parasites with a complex life cycle in urban environments).

Thank you for raising this interesting point. We have added this consideration to the Discussion, L324.

One question that arose while reading the text, which I wonder if the authors could elaborate on, is their selection of search criteria.

Thank you for raising this question. Our Methods now include much more detail on our selection of search criteria. We have addressed specific responses to your comments below.

This seems to be a critical part of their methods and in particular, the use of the search term 'disturbance' was a bit puzzling as it wasn't mentioned in the hypotheses and the logic for its inclusion was not immediately clear; in theory one could imagine abstracts comparing viruses of rural and urban rodents that need not mention 'disturbance' or 'parasite' at all. Did all of the search terms on L136 have to be present, or were papers that did not have the word 'disturbance' also included in their literature search?

We included the term disturbance because we encountered studies concerned with the effect of "human disturbance" on parasitism in mammals. This term was also used in a review by Chapman et al. (2005) and a meta-analysis by Young et al. (2013), both discussing the effect of habitat disturbance on parasitism in primates. Since "disturbance" is a blanket term that may encompass sites affected by urbanization as well as agriculture, logging, mining, etc., we felt we should substitute the word "urban" with this term to bring in papers we may have missed. We clarified this in L141. We also specified that we used the operator "OR", indicating that papers without the word "disturbance" were still included in the search.

It seems likely the authors explored several search term combinations to maximize the papers found and then settled on the most effective ones; perhaps the authors could simply describe their selection process/logic or explore these search parameters a bit more systematically in the text?

We understand that our previous wording- “we initially used... a combination of”- implied that there was some process of adding and substituting terms to make different combinations, but was too vague and therefore confusing. The text now specifies the exact terms and operators used to conduct each search, and clarifies the chronological sequence of these searches

For example for viruses, is ‘parasit*’ always included in the keywords, abstracts, or titles of papers that compare viral populations in mammal populations in rural and urban environments (e.g., while not comparing rural and urban populations explicitly, the paper “Non-random patterns in viral diversity” by Anthony et al., Nature Communications 2015 does not have the word parasite anywhere in the manuscript) and I wonder if the emphasis on parasites may have missed papers focusing on viruses or bacteria. It is certainly possible that efforts presented here have found most of the relevant papers on parasites of mammals in rural and urban settings, but by explaining their selection of search terms and being a bit clearer about the methods of the search, they might be able to convince the readers that this is the case. As a reality check, perhaps the authors could replace parasite with the search terms, “virus*”, “bacteria*”, “protozoa*”, or “helminth*”, and remove the word “disturbance” from their search terms to see whether any additional publications are discovered?

Thank you for raising this suggestion, which we followed by additional searches of the databases originally used. Replacing “parasit*” with “helminth*” or “protozoa*” or “virus*” or “bacteria*” uncovered three studies with new bacterial parasites, and one with a new viral parasite. By including these studies, we also added two new rodent host species. Our meta-analysis is therefore larger after adding these additional search terms and the resulting four publications.

Perhaps this sort of approach was what the authors were doing with their “genus-specific searchers” (L139-141), but what exactly was done was a bit unclear with the limited description provided and it did not seem to be sufficiently explained for an independent reproduction of their efforts. A more thorough and clear description of how papers were collected might be helpful.

We now list the genera that were included in our searches of the GMPD, as well as a reference discussing the presence of these genera in urban areas. We expect that inclusion of these terms will increase the replicability of our study.

Figure 1A: At least in the acrobat version used by the reviewer, the phylogeny seems to have been rendered with some issues – there is a vertical line to the left of the phylogeny that should be removed and the branches do not seem to connect in all cases; there are also some extra lines on some of the branches and there are some pieces of the label names that seem to have been over written that are still showing. Perhaps this was a problem in combining the two figures or is just a problem for the version of acrobat used by the reviewer (in which case please ignore this comment); it might be worth a double check that this figure is reproducing as planned. A scale bar for

the host phylogeny might be useful to add as well, to give the reader a sense of the deep evolutionary scales that were explored with this rich dataset.

We fine-tuned Figure 1A in Adobe Illustrator, and we expect this will solve any of the rendering issues. Given the importance of broad taxonomic classification in our analysis, but the lack of phylogenetic signal in residuals of the model, we opted to portray an unscaled phylogeny to primarily indicate the species included in our analysis and the relationships between them as a complement to our forest plot.

Figure 1B: It seems like it would be possible to indicate the host species in this figure; for example, the authors could color the host species for each group with a different color in A, and then color the log odds ratios in B accordingly. Alternatively, each host species could get a unique letter (or a unique capital or lowercase letter) or shape (e.g., a silhouette of the host), and this could be placed at the mean log odds ratio. In B, there are also some squares in some of the standard error bars (looks like they are at the mean log odds ratio) – this is also mentioned in the figure legend but I believe is no longer plotted for all parasites. This might be something to double check as it seems they should either be there for all of the log odds ratios, or removed for all. What would also be very helpful is if the authors could show the parasite life cycles in this figure at the same time as the host; it seems to be such a neat aspect of their study. While this potentially represents a lot of information, it could be done simply with the use of dashed or solid lines, or the use of capital and lowercase letters if the strategy mentioned above is used. The other option would be to create two of these forest plots, one for parasites with a simple life cycle, and one with those with a complex life cycle, which might improve interpretability. Given their striking finding regarding the impact of parasite life cycles on these log odds ratios, incorporating this information into this plot seems like it could be very helpful for the reader to understand the dataset. Perhaps in the figure, the authors could indicate what negative and positive values mean (i.e., higher parasite risk in rural areas <- -> higher parasite risk in urban areas).

We revised Figure 1B to represent a wealth of information as clearly as possible for the reader. Each mammalian order is more clearly delineated into separate sections. We followed the Referee's suggestion in representing host species within each section as a unique shape and color combination. We also represented parasite life cycle by dashed or dotted lines, and grouped effect sizes for simple and complex parasites together within each section for ease of interpretation. We also indicated within the figure that negative and positive log odds ratios correspond to lower and higher urban parasite prevalence.

Minor comments and suggestions:

Keywords: I wonder if epidemiology and zoonoses are appropriate key words given the changes the authors have made. Consider replacing.

Replaced “epidemiology” and “zoonoses” with “prevalence” and “transmission”. Also changed “parasitism” to “parasite” since the former is now in the title.

L21: Perhaps “potentially impact” rather than “profound consequences”, as that seems to be one the knowledge gap this article is addressing.

Changed to “potential consequences”.

L23: Perhaps rather than “use resources”, simply “survive” or “thrive”; there are potentially many factors that limit the ability of a species to thrive in an environment, not simply the use of resources.

Changed to “survive in urban settings”. The meta-analysis includes some species that do seem to thrive in urban habitat, and others that are found in urban habitat but do not necessarily thrive.

L40: Is this rate of expansion unprecedented in human history? On what scale (decades, years, centuries, millennia) and what metric do the authors have in mind (%land use, % of human population living in cities)? Consider rewording (e.g., urban areas are expanding and transforming landscapes) or clarifying what is unprecedented.

We opted to reword this and remove the term “unprecedented” rather than clarify so as not to distract from the emphasis on the current state of urbanization.

L54-56: I appreciate the sentiment, but I would encourage the authors to not put too negative a spin on urbanization, at least in terms of wildlife conservation. Indeed, there are many that argue that urbanization represents one way that humans could reduce their impact on wildlife populations; humans living in rural environments often are not leaving space for wildlife either (or humans only tolerate some kinds of wildlife in this rural areas), and a lot of close contact with wildlife (e.g., hunting) takes place that likely does not take place so much in urban settings. In addition, parasites are only one metric of an individual or an animal populations health or viability and parasites per say do not necessarily indicate a struggling individual or population. Given that only 3% of global land area falls into the category of ‘urban’, I think statements that urbanization is a rapid mechanisms of habitat loss leading to population decline are a bit extreme. I would encourage the authors to take a more nuanced perspective here and throughout the manuscript, on how urbanization might impact animal populations. This need not detract from their findings, which are striking and interesting. If people are leaving rural areas to live in cities, and these rural areas can, at least in part, be returned to wildlife, this could be a very efficient way to protect wildlife and reduce the human impact on wildlife populations. In addition, cities can be much more efficient for many services etc., so their environmental impact (e.g., carbon emissions) per capita may be less than for people living in rural areas. All that to say, the impact of urbanization on wildlife conservation is likely complicated.

Thank you for making this point, and we understand your concerns. Our goal is to focus on the expansion of urban landscapes, and to describe the shifts in community composition that occur in such landscapes, which relate directly to shifting host-parasite dynamics. We altered our introductory paragraph to begin with a broader, more neutral statement on urbanization that leaves room for consideration of nuanced effects on biodiversity. Then we state that the expansion of urban landscapes is one consequence of urbanization, and move forward with a focus on ecology within these landscapes. We also removed strong negative language towards urbanization throughout the Introduction. Through these modifications, we have retained our focus on urban ecology and taken a more open stance towards the overall impact of urbanization on wildlife.

L68: Consider removing the word profound and highlighting that this isn't always the case: "can" might be helpful here.

Done.

L91: Consider adding a "can" or "may"

Done.

L139-141: As discussed above, it was a bit unclear what was being done with these genus specific searches or how one might do this to replicate the study; please clarify. Should the database now be the GMPD rather than GPPD given the emphasis now on mammals?

Addressed in response above. We clarified in the L150 that only the primate database within the GMPD was included in the search because the primate database is in current working order, having been maintained and updated recently.

L307: Perhaps clarify with another word than taxa (could even list the bigger taxonomic groups meant)?

Changed to "mammalian orders"

L338: Did the authors confirm that these mammals are the definitive host for these parasites? It seemed a bit surprising that definitive was not mentioned in the intro, methods or results, but features in the discussion/conclusion; perhaps the authors could consider clarifying in the methods/results/discussion how this concept was brought into the analysis and interpretation or leave out this concept in the discussion and focus rather simply on complex vs. simple life cycles to simplify interpretation.

We no longer label primates and carnivores as definitive hosts in the concluding paragraph. This term now only appears in L277 when we discuss a specific

mechanism of disrupted parasite transmission that would apply to species that are definitive hosts.

L338-340: Given that the authors haven't looked at food abundance here and it hard to know how these parasites impact fitness and survival etc., this seems speculative. Consider removing here, and sticking to the really interesting findings uncovered through this meta-analysis.

Removed the speculation about abundance of food.

Referee: 1

Comments to the Author(s).

This is a much improved and expanded version of the previous manuscript. The distinction between urban exploiters and urban avoiders puts everything in a better context. There is a lot of synthesizing and categorizing the data in the discussion which was great, but I struggled to visualize it in the results; it might be nice to have a summary figure of the takeaway message. I've attached a very rough sketch (that you can take or leave) of how I saw it in my head. Overall, I think the authors did a nice job of bringing together a lot of studies that will hopefully spur more detailed (e.g. beyond genus ID) work from researchers in the future.

Thank you for your positive feedback, and thank you for including this example figure—it provided the basis for the summary figure that we created. We included a simple diagram with the basic takeaway points from our paper.

An editorial note: There is a font change from the end of line 224 through the end of 226.

Fixed

References:

Chapman CA, Gillespie TR, Goldberg TL. 2005 Primates and the ecology of their infectious diseases: How will anthropogenic change affect host-parasite interactions? *Evolutionary Anthropology: Issues, News, and Reviews* **14**, 134-144.

Young H, Griffin RH, Wood CL, Nunn CL. 2013 Does habitat disturbance increase infectious disease risk in primates? *Ecology Letters* **16**, 656-663.

Appendix C

09 April 2020

Dear Editor,

Thank you for accepting our manuscript, "The Effect of Urban Habitat Use on Parasitism in Mammals: A Meta-Analysis," for publication in *Proceedings of the Royal Society B*.

We appreciate all the feedback we have received from you and the referees, and we feel our manuscript has benefited greatly from the review process. Please see below for a response to the final modification suggested for our manuscript.

Thank you again, and we are honored to have our study published as an Evidence Synthesis Article in *Proceedings B*.

Sincerely,

Courtney S. Werner

Associate Editor Board Member: 1

Comments to Author:

Thank you for the thorough revision and for responding so constructively to the PRSB review process throughout. There is one remaining issue that I endorse, to enhance clarity of the Figure. We look forward to seeing this important MS published in the near future.

Reviewer(s)' Comments to Author:

Referee: 2

Comments to the Author(s)

I congratulate the authors for so thoroughly addressing all of the concerns and suggestions that arose through the review process. The modifications have clarified

all of my questions and concerns and the findings presented are interesting, compelling, and well presented.

One last minor suggestion; in Figures 1 and 2, the line type (dashed or solid) corresponding to parasites with a simple and complex lifecycle is switched. If it is not a major hassle, it might reduce potential confusion to modify one of the two figures so that the same line type is used for the corresponding life cycle types?

Thank you for working with us to clarify and improve the presentation of our findings. We agree that the line type corresponding to parasite life cycle type should be the same across Figure 1 and Figure 2. However, when we looked at the PDF proof of the manuscript submitted on April 02, we did not notice that the line type was switched between the two figures. Therefore, we did not alter either figure for this final submission. We will of course make clarifying changes if we missed something or misunderstood the referee's concerns.